# The lysophospholipase D enzyme Gdpd3 is required to maintain chronic myelogenous leukaemia stem cells

Kazuhito Naka [1✉], Ryosuke Ochiai [2], Eriko Matsubara [2], Chie Kondo [2], Kyung-Min Yang [3], Takayuki Hoshii [4], Masatake Araki [5], Kimi Araki [5], Yusuke Sotomaru [6], Ko Sasaki [7], Kinuko Mitani [7], Dong-Wook Kim [8], Akira Ooshima [3] & Seong-Jin Kim [3,9,10]

Although advanced lipidomics technology facilitates quantitation of intracellular lipid components, little is known about the regulation of lipid metabolism in cancer cells. Here, we show that disruption of the *Gdpd3* gene encoding a lysophospholipase D enzyme significantly decreased self-renewal capacity in murine chronic myelogenous leukaemia (CML) stem cells in vivo. Sophisticated lipidomics analyses revealed that *Gdpd3* deficiency reduced levels of certain lysophosphatidic acids (LPAs) and lipid mediators in CML cells. Loss of *Gdpd3* also activated AKT/mTORC1 signalling and cell cycle progression while suppressing Foxo3a/β-catenin interaction within CML stem cell nuclei. Strikingly, CML stem cells carrying a hypomorphic mutation of *Lgr4/Gpr48*, which encodes a leucine-rich repeat (LRR)-containing G-protein coupled receptor (GPCR) acting downstream of Gdpd3, displayed inadequate disease-initiating capacity in vivo. Our data showing that lysophospholipid metabolism is required for CML stem cell maintenance in vivo establish a new, biologically significant mechanism of cancer recurrence that is independent of oncogene addiction.

[1] Department of Stem Cell Biology, Research Institute for Radiation Biology and Medicine, Hiroshima University, 1-2-3, Kasumi, Minami-ku, Hiroshima 734-8553, Japan. [2] Pharmaceuticals and Life Sciences Division, Shimadzu Techno-Research, Inc., 1, Nishinokyo-shimoai-cho, Nakagyou-ku, Kyoto 604-8436, Japan. [3] Precision Medicine Research Center, Advanced Institute of Convergence Technology, Seoul National University, #C-504, 145, Gwanggyo-ro, Yeongtong-gu, Suwon, Gyeonggi-do 16229, Republic of Korea. [4] Department of Molecular Oncology, Graduate School of Medicine, Chiba University, 1-8-1, Inohana, Chuo-ku, Chiba 260-8670, Japan. [5] Institute of Resource Development and Analysis, Kumamoto University, 2-2-1, Honjo, Chuo-ku, Kumamoto 860-0811, Japan. [6] Natural Science Center for Basic Research and Development, Hiroshima University, 1-2-3, Kasumi, Minami-ku, Hiroshima 734-8551, Japan. [7] Department of Hematology and Oncology, Dokkyo Medical University School of Medicine, 880, Kitakobayashi, Mibu, Shimotsuga-gun, Tochigi 321-0293, Japan. [8] Catholic Hematology Hospital, Leukemia Research Institute, The Catholic University of Korea, #222 Banpo-daero, Seocho-gu, Seoul 137-701, Republic of Korea. [9] Department of Transdisciplinary Studies, Graduate School of Convergence Science and Technology, Seoul National University, #C-504, 145, Gwanggyo-ro, Yeongtong-gu, Suwon, Gyeonggi-do 16229, Republic of Korea. [10] MedPacto Inc., 92, Myeongdal-ro, Seocho-gu, Seoul 06668, Republic of Korea. ✉email: kanaka55@hiroshima-u.ac.jp

Glycerophospholipids (phospholipids) organise the lipid bilayer in a cell's plasma membrane. These phospholipids are also a source of lipid mediators such as prostaglandins and leukotrienes, which play essential roles in immune responses, inflammation, and cancer development[1–3]. Phospholipids contain two hydrophobic fatty acid chains at the sn-1 and sn-2 sites, and one hydrophilic polar base (choline, serine, inositol or ethanolamine) at the sn-3 site (Fig. 1a, upper right). Since the 1950s, it has been known that glycerol 3-phosphate (G3P) is first converted into lysophosphatidic acids (LPAs) and then into various phospholipids via the Kennedy pathway (the so-called de novo pathway)[4], as illustrated in Fig. 1a. To supply a wide variety of lipid molecules, the fatty acid chains and polar bases of phospholipids undergo exchange via the Lands' cycle (remodelling pathway) to create distinct types of lysophospholipids[5,6]. These lysophospholipids can then be recycled back into LPAs through the activities of three lysophospholipase D enzymes, namely Autotaxin (ATX), GDPD3 (also termed GDE7), and GDPD1 (GDE4), which have the ability to specifically hydrolyse the polar base at the sn-3 site of a lysophospholipid[7–9].

Lysophospholipids have only one fatty acid chain at the sn-1 or sn-2 site and so are more water-soluble than phospholipids, allowing them to act as lipid second messengers mediating regulatory signalling. These molecules convey their signalling functions by binding to certain LPA receptors (LPAR1-LPAR6) belonging to the G-protein coupled receptor (GPCR) family[10,11]. However, the precise roles of lysophospholipid metabolism in cancer cells in general, and in cancer stem cells in particular, are not yet understood. It is known that ATX-disrupted mice exhibit embryonic lethality, and that enforced expression of ATX or LPARs promotes the initiation and metastasis of breast cancer in transgenic mouse strains[12,13]. In humans, our knowledge is even more limited, although aberrant expression of phosphatidylcholines (PCs) has been detected in patients with colorectal cancer, triple-negative breast cancer, or lung adenocarcinoma[14–16].

Chronic myelogenous leukaemia (CML) stem cells are the hierarchal apex cells in CML disease. CML stem cells have both the capacity to self-renew and to produce mature CML cells[17]. Despite their expression of the BCR-ABL1 oncogene, CML stem cells have been reported to maintain their stemness in an oncogene-independent manner[18], the mechanism of this maintenance is unknown. Thus, although the advent of tyrosine kinase inhibitors (TKIs) has dramatically improved the prognoses of CML patients, CML stem cells are untouched by TKI treatment and survive to cause the relapse of CML disease[19]. A cure for CML therefore remains elusive.

The oncogene-independent survival of CML stem cells has spurred many researchers to search for CML stem cell-specific vulnerabilities in the metabolic pathways controlling their energy production, amino acid acquisition, and lipid mediator generation[20]. For instance, activation of the PPARγ-mediated signalling pathway by its agonist pioglitazone can reduce CML stem cells in human patients[21]. Among enzymes involved in lipid metabolism, arachidonate 5-lipoxygenase (Alox5) and arachidonate 15-lipoxygenase (Alox15) are known to be essential for CML stem cell survival[22,23]. When used in combination with the TKI imatinib, prostaglandin $E_1$ (PGE$_1$) can reduce relapse frequency in CML-affected mice[24]. We previously reported that forkhead O transcription factor 3a (Foxo3a), which is regulated by phosphatidyl-inositol 3-phosphokinase (PI3K) and AKT, plays a crucial part in controlling CML stem cell function[25]. However, it has been difficult to pin down the biological role of lipidogenesis in the maintenance of CML stem cells.

In this study, we show that the Gdpd3 gene encoding a lysophospholipase D enzyme is more highly expressed in murine CML stem cells than in normal wild-type (WT) haematopoietic stem cells (HSCs). Most importantly, genetically Gdpd3-deficient CML stem cells are impaired in their long-term (LT) self-renewal capacity after serial transplantation. In addition, the mutant cells exhibit a reduced ability to cause disease relapse in TKI-treated recipient mice, indicating that Gdpd3-mediated lysophospholipid metabolism is a major contributor to the maintenance of CML stem cells in vivo. Our results also demonstrate that the lysophospholipid pathway sustains the production of lipid mediators and the expression of GPCR genes (including Lgr4/Gpr48) that are required for CML stem cell function in vivo. Thus, our work has uncovered and linked two features of CML stemness: (1) the lysophospholipase D Gdpd3 is critical for maintaining primitive CML stem cells in an oncogene-independent manner, and (2) lysophospholipid metabolism produces vital lipid mediators and regulates multiple downstream pathways, including GPCR-mediated signalling, that are essential for CML stem cell function.

## Results

**Gdpd3 contributes to CML stem cell survival in vivo.** To identify lipid metabolism-related genes that are differentially expressed in CML stem cells compared to normal WT HSCs, we surveyed an RNA-Seq gene expression profiling dataset that we previously made available to a public database gene expression omnibus (GEO, ID: GSE70031, NCBI, NIH, USA)[26]. We found that the Gdpd3 gene encoding a lysophospholipase D enzyme was more highly expressed in the most primitive LT-CML stem cells than in normal WT LT-HSCs (Supplementary Fig. 1). This finding prompted us to investigate the biological significance of Gdpd3 and lysophospholipid metabolism in CML stem cells. For this study, we used two CML mouse models: (1) SCL-tTA x TRE-BCR-ABL1 double transgenic CML mice, the so-called tet-inducible CML-affected mouse model[27,28], designated herein as tet-CML mice; and (2) the retroviral BCR-ABL1 transduction CML model, termed the retro-CML-affected mouse model, designated herein as retro-CML mice. The latter mutants were derived by bone marrow transplantation (BMT) of murine HSCs that were retrovirally transduced with the BCR-ABL1-ires-EGFP gene, as reported in our earlier study[25,26]. The tet-CML model is best suited for evaluating the natural development of CML disease in mice, and for examining molecular mechanisms in the most primitive LT-CML stem cells. On the other hand, retro-CML mice are very useful for examining the functionality (i.e., maintenance of CML disease-initiating capacity in vivo) of CML-LSK (Lineage$^-$Sca-1$^+$cKit$^+$) cells during serial BMT.

We first harvested bone marrow mononuclear cells (BMMNCs) from littermate healthy control mice (SCL-tTA$^+$) and tet-CML mice (SCL-tTA$^+$TRE-BCR-ABL1$^+$), and isolated LT-stem cells (CD150$^+$CD48$^-$CD135$^-$LSK), CD48$^+$LSK cells, multipotent progenitors (MPP), and LK (Lineage$^-$Sca-1$^-$cKit$^+$) populations (Supplementary Fig. 2). Application of real-time PCR analysis to these cells revealed that the Gdpd3 gene was more highly expressed in LT-stem cells and CD48$^+$ LSK cells from tet-CML mice compared to the same populations from healthy littermate control mice, but that the same was not true for MPP and LK cells (Fig. 1b). Notably, the transduction of either of two siRNAs targeting mouse Gdpd3 mRNA suppressed the colony-forming capacity of CML-LSK cells (Fig. 1c). This particular gene's candidacy as a key lipid metabolism enzyme contributing to CML was also bolstered by the results of a preliminary lipidomics analysis (see below).

To examine Gdpd3's role in CML in vivo, we used genome-editing technology in mouse embryos to successfully establish a Gdpd3-deficient mouse strain (Gdpd3$^{-/-}$) in which five bp between nucleotides 15 and 19 after the first methionine of the Gdpd3 gene were deleted (Supplementary Fig. 3). Consistent with

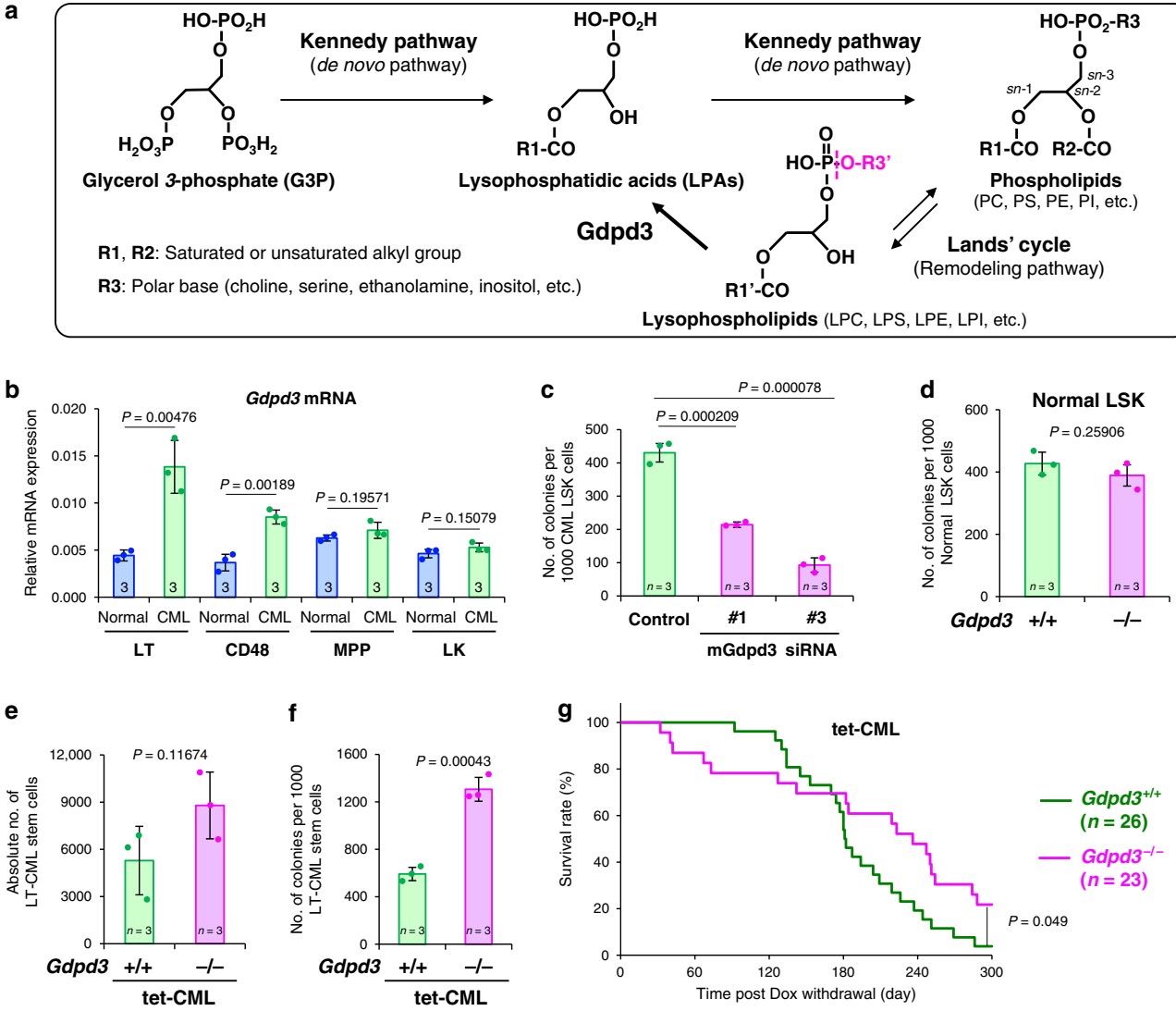

**Fig. 1 Gdpd3 is implicated in CML disease initiation in vivo. a** Diagram of pathways of lysophospholipid biosynthesis. G3P is converted into LPAs, and LPAs are then converted into phospholipids by the addition of polar bases via the Kennedy (de novo) pathway. The Lands' cycle (remodelling pathway) generates lysophospholipids of distinct composition by substituting fatty acid ester and polar base groups of phospholipids. Lysophospholipase D Gdpd3 converts lysophospholipids back into LPAs by catalysing hydrolysis (magenta dotted line). (PC Phosphatidylcholine, PS Phosphatidylserine, PE Phosphatidylethanolamine, PI Phosphatidylinositol, LPC Lysophosphatidylcholine, LPS Lysophosphatidylserine, LPE Lysophosphatidylethanolamine, LPI Lysophosphatidylinositol). **b** qRT-PCR determination of *Gdpd3* mRNA expression in LT-stem (LT), CD48, MPP, and LK cells (see Supplementary Fig.2) isolated from *Gdpd3*$^{+/+}$tet-CML-affected (SCL-tTA$^+$TRE-BCR-ABL1$^+$) mice (one male, six females) or normal littermate (SCL-tTA$^+$) mice (four males, four females). Data are the mean ratio ± s.d. of transcript levels normalised to *Actb* ($n = 3$ biologically independent samples) (*P*-value, unpaired two-sided Student's *t*-test). **c** Quantitation of the colony-forming capacity of *Gdpd3*$^{+/+}$ CML-LSK cells that were transduced with/without Cy3-labelled siRNA targetting mouse *Gdpd3* mRNA (mGdpd3 siRNA #1 or #3). Cy3$^+$ and Cy3$^-$ CML-LSK cells were purified at 3 days post-transduction and plated in semi-solid methylcellulose medium. Data are the mean colony number ± s.d. ($n = 3$) and are representative of biologically independent three experiments (*P*-value compared with control, unpaired two-sided Student's *t*-test). **d** Quantitation of colony-forming capacity of haematopoietic stem/progenitor (LSK) cells that were isolated from normal *Gdpd3*$^{+/+}$ (one female) and *Gdpd3*$^{-/-}$ (one female) mouse and cultured in semi-solid methylcellulose medium under hypoxic (3% O$_2$) conditions. Data are the mean colony number±s.d. ($n = 3$) and are representative of biologically independent two experiments (*P*-value, unpaired two-sided Student's *t*-test). **e** Absolute numbers of LT-CML stem cells isolated from the two hind limbs of *Gdpd3*$^{+/+}$tet-CML-affected mice (three females, $n = 3$ biologically independent samples) and *Gdpd3*$^{-/-}$tet-CML-affected mice (one male, two females, $n = 3$ biologically independent samples). Data are the mean absolute numbers ± s.d. of LT-CML stem cells (*P*-value, unpaired two-sided Student's *t*-test). **f** Quantitation of colony-forming capacity of LT-CML stem cells that were isolated from a *Gdpd3*$^{+/+}$tet-CML-affected mouse (one male) and a *Gdpd3*$^{-/-}$tet-CML-affected mouse (one male) and co-cultured on an OP-9 stromal cell layer under hypoxic (3% O$_2$) conditions. Data are the mean colony number±s.d. ($n = 3$) and are representative of biologically independent three experiments (*P*-value, unpaired two-sided Student's *t*-test). **g** Survival rates of *Gdpd3*$^{+/+}$tet-CML-affected mice (18 males, eight females, $n = 26$ biologically independent samples) and *Gdpd3*$^{+/+}$tet-CML-affected mice (19 males, four females, $n = 23$ biologically independent samples) after induction of CML by Dox withdrawal (*P*-value, Log-rank non-parametric test).

a previously reported Gdpd3-deficient mutant[29], our Gdpd3−/− mice were born at the expected Mendelian ratio and appeared healthy (data not shown). Thus, the Gdpd3-mediated lysophospholipid metabolic pathway is dispensable for normal mouse development and survival. With respect to haematopoesis, whereas the red blood cell (RBC) count and haematocrit (HCT) appeared to increase in Gdpd3−/− mice compared to Gdpd3+/+ mice, no difference was observed in white blood cell (WBC) count (Supplementary Table 1). In addition, no differences were observed in CFU-GM, BFU-E, or CFU-mix populations of BMMNCs, or in the colony-forming capacity of LSK cells isolated from normal healthy Gdpd3+/+ and Gdpd3−/− littermate mice (Fig. 1d; Supplementary Fig. 4). In serial BMT experiments, normal LSK cells from Gdpd3−/− mice maintained their BM reconstitution capacity through the first- and second-rounds of such transplantation in vivo (Supplementary Fig. 5).

We next employed our Gdpd3−/− tet-CML mouse model to evaluate Gdpd3's function in CML stem cells. The absolute number of LT-CML stem cells was modestly higher in Gdpd3−/− tet-CML-affected mice than in Gdpd3+/+ tet-CML controls (Fig. 1e). Unexpectedly, Gdpd3−/− LT-CML stem cells displayed greater colony-forming capacity than Gdpd3+/+ LT-CML stem cells (Fig. 1f). Consistent with this finding, Gdpd3−/− tet-CML mice developed CML disease more rapidly than Gdpd3+/+ tet-CML mice, usually within three months of CML induction by Dox withdrawal (Fig. 1g). However, this enhanced disease-initiating capacity had attenuated by six months post-Dox withdrawal such that more Gdpd3−/− tet-CML mice survived longer than did Gdpd3+/+ tet-CML mice. These results suggest that, even though the loss of Gdpd3 in CML stem cells initially aggravates CML disease, these cells eventually lose the ability to differentiate into mature CML cells.

To examine the self-renewal capacity of Gdpd3-deficient CML-LSK cells in vivo, we employed our retro-CML model mice[25]. Consistent with the accelerated initiation of CML disease observed in our tet-CML model, recipient mice transplanted with Gdpd3−/− retro-CML-LSK cells developed CML disease more rapidly than recipients transplanted with Gdpd3+/+ retro-CML-LSK cells (Fig. 2a). However, to our surprise, serial BMT experiments indicated that Gdpd3−/− retro-CML-LSK cells isolated from primary recipients showed a marked decrease in disease-initiating capacity in second-round recipients compared to Gdpd3+/+ retro-CML-LSK cells (Fig. 2b). We next examined the effect of Gdpd3 loss on the absolute number and frequency of BCR-ABL1/EGFP+ CML-LSK cells in first- and second-round BMT recipients. No significant differences were observed in either the frequency or the absolute number of BCR-ABL1/EGFP+ CML-LSK cells in BM of first-round recipients (Fig. 2c; Supplementary Fig. 6a). However, in spleen, the absolute number of CML-LSK cells was dramatically increased in first-round recipients, even though we could not detect a difference in the frequency of these cells (Fig. 2d; Supplementary Fig. 6b). This elevated absolute number of CML-LSK cells in spleen resulted from a higher number of total splenocytes in Gdpd3−/− retro-CML mice compared to Gdpd3+/+ retro-CML mice (data not shown). When we examined second-round recipients, however, we found that CML-LSK cells were dramatically decreased in both BM and spleen of recipients bearing the mutant cells (Fig. 2e, f; Supplementary Fig. 6c, d). These data prompted us to compare the cell cycle distribution of Gdpd3+/+ and Gdpd3−/− CML-LSK cells isolated from retro-CML mice subjected to in vivo BrdU incorporation assays. The frequency of BrdU+ S phase cells in the spleen was strikingly elevated among Gdpd3−/− CML-LSK cells compared to Gdpd3+/+ CML-LSK cells (Fig. 2g, h), with a similar trend in BM (Supplementary Fig. 7). Accordingly, the frequency of BrdU− $G_0/G_1$ cells among Gdpd3−/− CML-LSK cells

was reduced compared to Gdpd3+/+ CML-LSK cells. These results demonstrate that Gdpd3 deficiency promotes the cell cycle progression of CML stem/progenitor cells, and thus breaks the quiescence essential for maintaining the stemness of CML stem cells in vivo.

**LPAs and lipid mediators are decreased in Gdpd3−/− CML cells.** Given that the Gdpd3 gene encodes a lysophospholipase D enzyme that hydrolyses lysophospholipids into LPAs[8,9], we investigated levels of various LPAs predicted to be products of Gdpd3 activity in BM cells of normal WT, Gdpd3+/+ tet-CML and Gdpd3−/− tet-CML-affected mice. Interestingly, an examination of normal WT BMMNCs and CML BM cells showed that the presence of CML disease dramatically altered the levels of many LPAs (Fig. 3a). Several LPAs, including LPA20:4 (arachidonate), were increased in Gdpd3+/+ CML BM cells compared to normal WT BMMNCs, whereas only LPA18:1 (oleate) was decreased in this experiment. In contrast, most of these same LPAs tended to decrease in both BMMNCs and LSK cells isolated from Gdpd3−/− tet-CML mice compared to those from Gdpd3+/+ tet-CML mice (Fig. 3a, b). Thus, in general, loss of Gdpd3 enzymatic activity decreases LPA levels in the CML context.

We next performed a sophisticated global lipidomics analysis of 196 lipid mediators and discovered that several non-LPA lipid mediators were also markedly decreased in Gdpd3−/− tet-CML BM cells compared to Gdpd3+/+ tet-CML BM cells (Fig. 4). Notably, prostaglandin $D_2$ (PGD$_2$) and PGE$_2$, which were increased in Gdpd3+/+ tet-CML BM cells compared to normal WT BMMNCs, appeared to be decreased in Gdpd3−/− tet-CML BM cells compared to Gdpd3+/+ tet-CML BM cells (Fig. 4a). In the same vein, levels of eicosanoid and docosanoid fatty acids, such as 5-oxo-6E,8Z,11Z,14Z-eicosatetraenoic acid (5-KETE), 15-hydroxy-5Z, 8Z,11Z,13E,17Z-eicosapentaenoic acid (15-HEPE), and 17-hydroxydocosahexaenoic acid (17-HDHA), tended to be lower in Gdpd3−/− tet-CML BM cells than in Gdpd3+/+ tet-CML BM cells (Fig. 4b). Lyso-PAF (platelet-activating factor) was higher in Gdpd3+/+ tet-CML BM cells than in normal WT BMMNCs, but decreased in Gdpd3−/− tet-CML BM cells compared to Gdpd3+/+ tet-CML BM cells (Fig. 4c). Taken together, these data suggest that Gdpd3 deficiency decreases many important lipid mediators in CML cells in vivo.

**Loss of Gdpd3 activates the AKT/mTORC1 pathway.** To understand how Gdpd3 deficiency affects downstream signalling pathways, we first investigated the AKT/mTORC1 pathway in LT-CML stem cells isolated from Gdpd3+/+ and Gdpd3−/− tet-CML mice. Interestingly, levels of phosphorylated AKT and S6 ribosomal protein were increased in Gdpd3−/− LT-CML stem cells compared to Gdpd3+/+ LT-CML stem cells, indicating that Gdpd3 acts to suppress the AKT/mTORC1 pathway in Gdpd3+/+ LT-CML stem cells (Fig. 5a, b; Supplementary Fig. 8a, b). We then examined the subcellular localisation of Foxo3a, whose export from a cell's nucleus to its cytoplasm is induced by activated AKT. We observed that, while Foxo3a was located within the nuclei of Gdpd3+/+ LT-CML stem cells as expected[25], Foxo3a was predominantly detected in the cytoplasm of Gdpd3−/− LT-CML stem cells (Fig. 5c; Supplementary Fig. 8c). These data suggest that the increased level of activated AKT/mTORC1 signalling occurring in the absence of Gdpd3 suppresses Foxo3a import into the nucleus in primitive LT-CML stem cells in vivo.

**TKI treatment is more effective in the absence of Gdpd3.** Although the discovery of TKIs has dramatically improved the prognoses of CML patients, the insuperable problem remains that

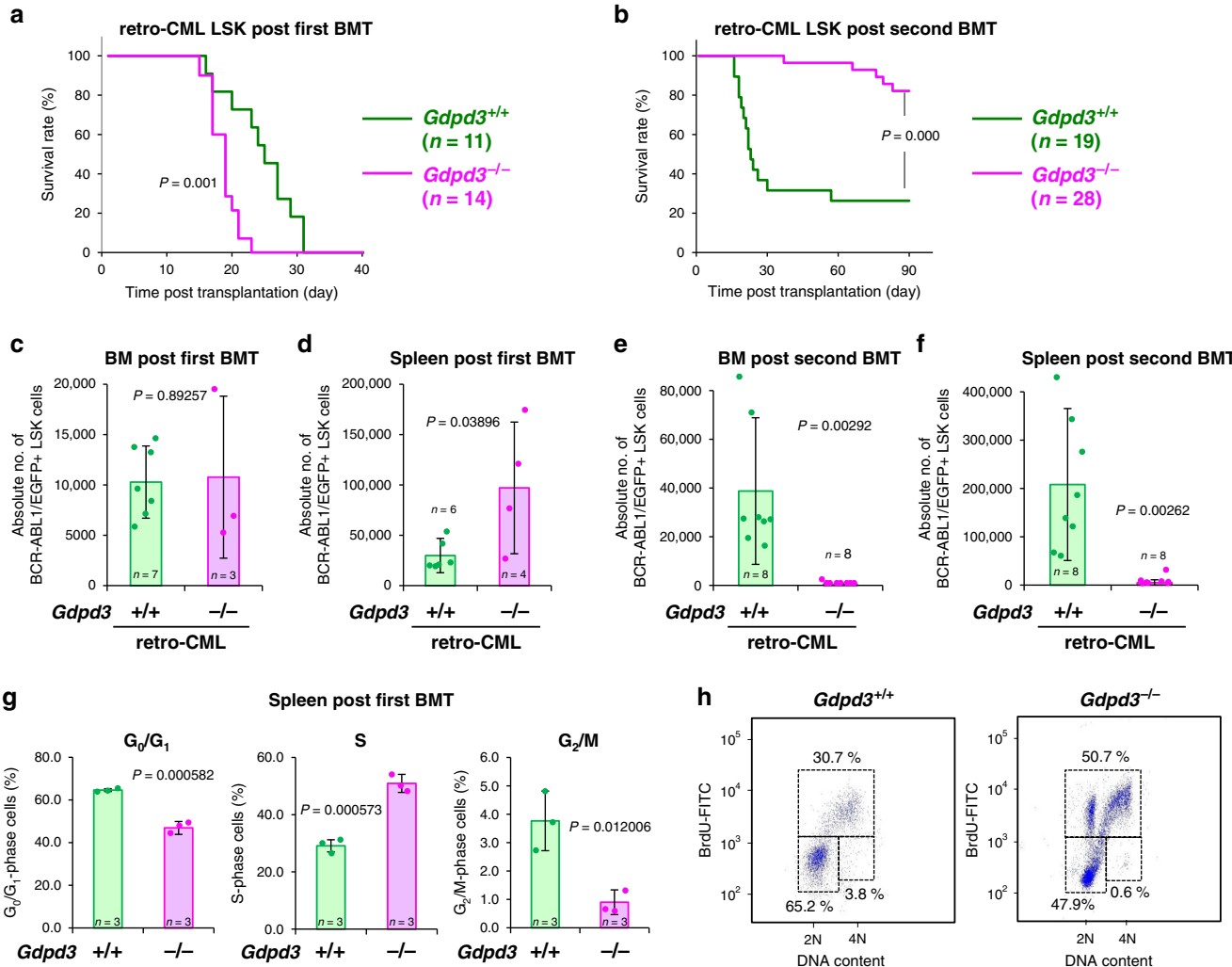

**Fig. 2 Gdpd3 maintains the self-renewal capacity of CML stem cells in vivo. a** Survival rates of retro-CML-affected mice in a primary BMT experiment that received BCR-ABL1/EGFP-transduced LSK cells from either *Gdpd3*[+/+] mice (one male, one female) or *Gdpd3*[−/−] mice (two females). Results shown are cumulative data obtained from two biologically independent experiments (*P*-value, Log-rank non-parametric test). **b** The survival rate of second-round BMT recipient mice that received serial transplantation of BCR-ABL1/EGFP[+] CML-LSK cells ($3 \times 10^4$ cells) isolated from either the *Gdpd3*[+/+] retro-CML-affected mice (14 females) or *Gdpd3*[−/−] retro-CML-affected mice (15 females) as in **a**. Results shown are cumulative data obtained from three biologically independent experiments. Mouse survival was monitored for 90 days. (*P*-value, Log-rank non-parametric test) **c–f** Absolute numbers of BCR-ABL1/EGFP[+] CML-LSK cells isolated from **c**, **e** BM of the two hind limbs and **d**, **f** spleen of *Gdpd3*[+/+] retro-CML-affected mice and *Gdpd3*[−/−] retro-CML-affected mice after **c**, **d** a first-round or **e**, **f** second-round of serial BM transplantation. Data are the mean absolute numbers ± s.d. of BCR-ABL1/EGFP[+] LSK cells. n numbers in **a**–**f** indicate biologically independent mouse numbers. (*P*-value, unpaired two-sided Student's *t*-test). (See also Supplementary Fig. 6). **g**, **h** Cell cycle distribution of CML-LSK cells in the spleen of *Gdpd3*[+/+] retro-CML-affected mice (three females) or *Gdpd3*[−/−] retro-CML-affected mice (three females) that were intraperitoneally administered BrdU for 3 h after a first-round of BMT as in **a**. Results are the mean frequency ± s.d. of $G_0/G_1$ phase CML-LSK cells, BrdU[+] S-phase CML-LSK cells, and $G_2/M$ phase CML-LSK cells ($n = 3$ biologically independent samples) (*P*-value, unpaired two-sided Student's *t*-test). (See also Supplementary Fig. 7). **h** Representative dot plots of BrdU incorporation and DNA content for the CML-LSK cells in **g**.

TKI therapy does not kill the CML stem cells responsible for disease relapse. To determine whether Gdpd3-mediated lyso-phospholipid metabolism was necessary for disease-relapsing capacity in CML stem cells, we transplanted recipient mice with *Gdpd3*[+/+] or *Gdpd3*[−/−] retro-CML-LSK cells in a first-round of BMT and treated the animals with the TKI dasatinib. Disease relapse was significantly decreased in treated recipients bearing *Gdpd3*[−/−] retro-CML-LSK cells compared to those that received *Gdpd3*[+/+] retro-CML-LSK cells (Fig. 6a). Thus, Gdpd3 expression in CML stem cells confers resistance to TKI therapy.

To determine if this increased survival of BMT recipients bearing *Gdpd3*[−/−] retro-CML-LSK cells was due to an inability to produce mature leukaemia cells, we used flow cytometry to examine the frequency of BCR-ABL1/EGFP[+] mature leukaemia cells in peripheral blood (PB). At 16–24 days post-BMT, we found no apparent differences in leukaemia cell numbers between *Gdpd3*[+/+] retro-CML-affected and *Gdpd3*[−/−] retro-CML-affected mice that had not been treated with dasatinib (Fig. 6b). In contrast, at 40–60 days post-BMT in mice treated with dasatinib, the frequency of BCR-ABL1/EGFP[+] leukaemia cells was significantly decreased in PB of *Gdpd3*[−/−] retro-CML-affected mice compared to PB of *Gdpd3*[+/+] retro-CML-affected mice (Fig. 6c). Flow cytometry following a t-SNE (t-Distributed Stochastic Neighbour Embedding) algorithm analysis indicated that PB samples from untreated control and *Gdpd3*-deficient retro-CML-affected mice both contained BCR-ABL1/EGFP[+]

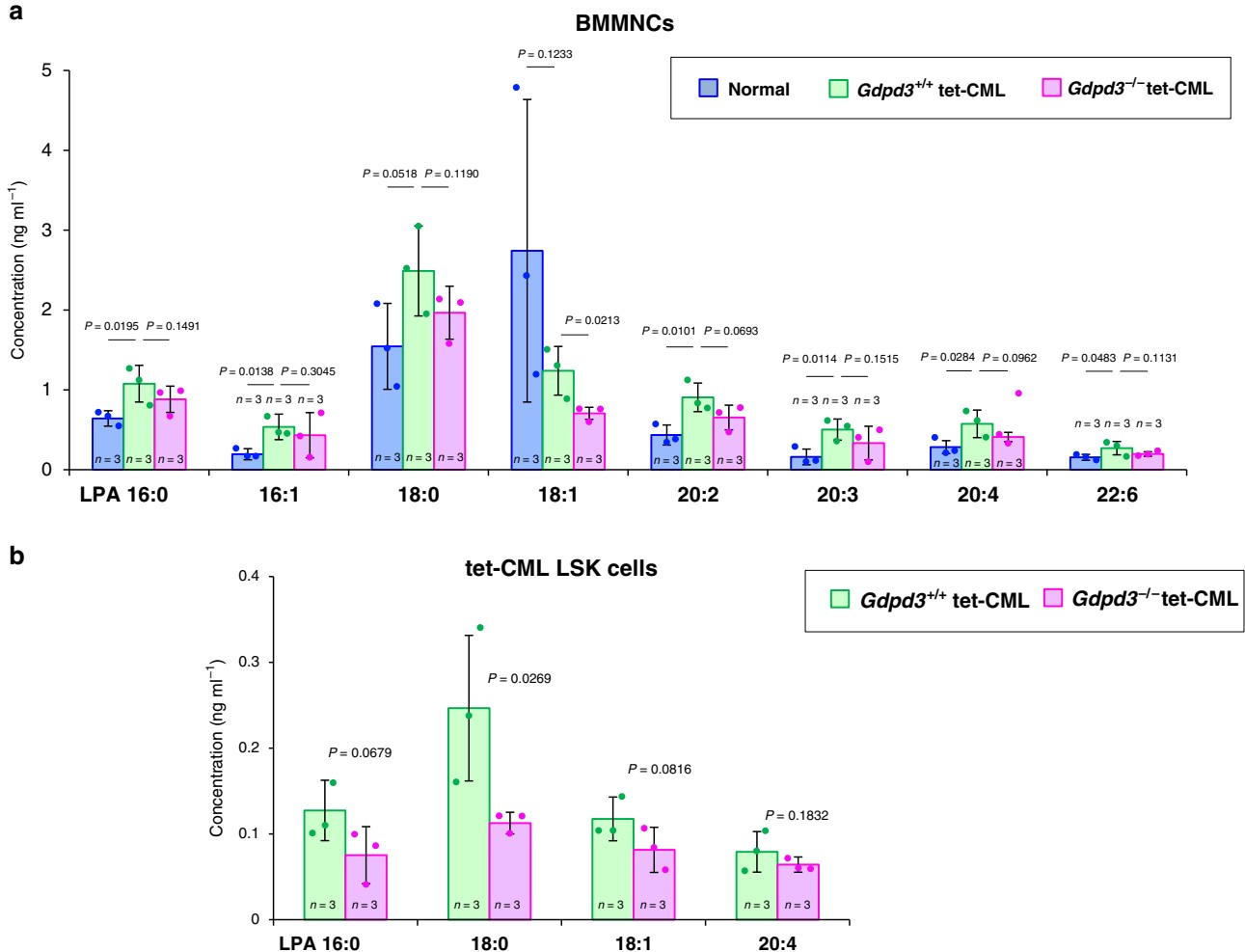

**Fig. 3 The loss of the *Gdpd3* gene decreases LPAs in CML cells in vivo. a**, **b** Global lipidomics analyses (see Methods and Supplementary Method 1) for the indicated LPAs in **a** BMMNCs ($1 \times 10^7$) isolated from healthy 8-wk-old normal C57BL/6 mice (two males, four females), *Gdpd3*$^{+/+}$ tet-CML-affected mice (five females), or *Gdpd3*$^{-/-}$ tet-CML-affected mice (one male, three females), and in **b** CML-LSK cells ($5 \times 10^5$) isolated from *Gdpd3*$^{+/+}$ tet-CML-affected mice (four males and 11 females), or *Gdpd3*$^{-/-}$ tet-CML-affected mice (six males, ten females), at 5 weeks post-Dox withdrawal. Data are the mean concentration (ng ml$^{-1}$) ±s.d. (*n* = 3 biologically independent samples) (*P*-value, unpaired one-sided Student's *t*-test). The *X*-axis labels represent LPA [carbon number]:[unsaturated bound number] of the fatty acid chain in the LPA.

leukaemia cells that were distinct from the populations of normal myelogenous cells, T cells and B cells in PB of normal healthy WT mice (Supplementary Fig. 9a, b, e). Thus, *Gdpd3*$^{-/-}$ retro-CML-LSK cells do have the ability to give rise to BCR-ABL1/EGFP-expressing mature leukaemia cells. However, although BCR-ABL1/EGFP$^+$ leukaemia cells were still present in PB of dasatinib-treated *Gdpd3*$^{+/+}$ retro-CML-affected mice (Supplementary Fig. 9c), they were dramatically decreased in PB from dasatinib-treated *Gdpd3*$^{-/-}$ retro-CML-affected mice (Supplementary Fig. 9d). Thus, in the absence of Gdpd3, TKIs are more effective in reducing disease relapse caused by CML stem cells.

To investigate whether *GDPD3* expression was altered in human CML patients, we retrieved data on levels of *GDPD3* mRNA in cells of imatinib-treated CML patients listed in a public database GEO (GEO: GSE12211)[30]. Intriguingly, for five out of the six CML patients examined, *GDPD3* mRNA levels were indeed higher in CD34$^+$ cells isolated after imatinib therapy than in CD34$^+$ cells isolated before treatment (Fig. 6d). These results implicate GDPD3 in the survival of human CML stem/progenitor cells.

We next examined the in vitro effects of siRNA targeting of human GDPD3 mRNA in the K562 human CML cell line as well as in primary human CD34$^+$ CML cells isolated from BM of a chronic phase CML patient. We transduced K562 cells or primary CD34$^+$ CML cells with FITC-labelled hGDPD3 siRNA#1 or #2 and purified cells containing FITC-hGDPD3 siRNAs by cell sorting (Supplementary Fig. 10). Although siRNA-mediated repression of hGDPD3 mRNA decreased the colony-forming capacity of K562 cells in vitro, the suppressive effect was only subtle (Fig. 6e). In contrast, the colony-forming capacity of human CD34$^+$ cells transduced with hGDPD3 siRNA was quite dramatically reduced (Fig. 6f). When we repeated this experiment in the presence of imatinib, we observed some colony formation by imatinib-resistant CD34$^+$ cells that had not been transduced with hGDPD3 siRNA. Intriguingly, the combination of imatinib plus hGDPD3 siRNA effectively suppressed colony formation by these resistant cells (Fig. 6g). These results reinforce our hypothesis that GDPD3 may contribute to the maintenance of primitive CML cells (rather than to the differentiation of mature CML cells) in humans as well as in mice.

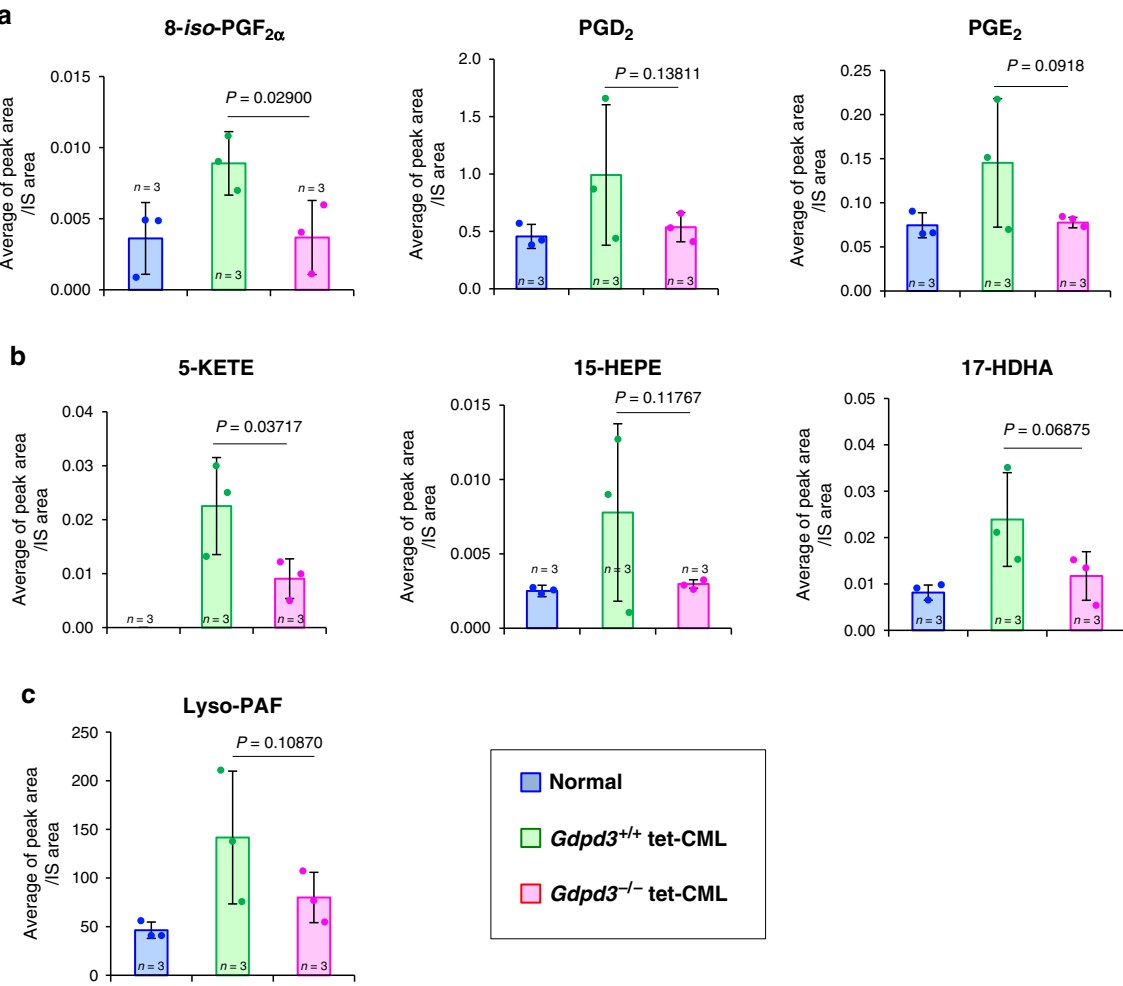

**Fig. 4 Lysophospholipase D Gdpd3 is required for the production of lipid mediators in CML cells in vivo.** Global lipidomics analyses (see "Methods", Supplementary Method 2, 3) for the lipid mediators in BMMNCs ($1 \times 10^7$) isolated from healthy 8-wk-old normal C57BL/6 mice (two males, four females), or $Gdpd3^{+/+}$ tet-CML-affected mice (five females), or $Gdpd3^{-/-}$ tet-CML-affected mice (one male, three females), at 5 weeks post-Dox withdrawal. Data are the mean average±s.d. of peak area normalised to IS (internal standard) area ($n = 3$ biologically independent samples) (P-value, unpaired one-sided Student's t-test). **a** Prostanoids, **b** eicosanoids and docosanoid, and **c** Lyso-PAF were indicated.

**Loss of *Gdpd3* results in aberrant expression of GPCR genes.** To understand how *Gdpd3* deficiency impairs the self-renewal capacity of CML stem cells in vivo, we compared gene expression profiles of murine $Gdpd3^{-/-}$ LT-CML stem cells and $Gdpd3^{+/+}$ LT-CML stem cells. RNA-Seq followed by gene ontology (GO) term enrichment analyses indicated that several GPCR family genes, including *Lpar4/Gpr23*, *Ltb4r1/Gpr16*, *Gpr82*, and *Gpr84*, were decreased in $Gdpd3^{-/-}$ LT-CML stem cells compared to $Gdpd3^{+/+}$ LT-CML stem cells (Fig. 7a; Supplementary Fig. 11a, b). Interestingly, *Lpar4/Gpr23* encodes a GPCR that binds to LPAs which are the predicted products of the Gdpd3 enzyme[31]. Thus, a defect in lysospholipid hydrolysis due to loss of Gdpd3 activity might affect *Lpar4/Gpr23* expression. Strikingly, we found that mRNA levels of *Lgr4/Gpr48*, which encodes a leucine-rich repeat (LRR)-containing GPCR, were also significantly lower in $Gdpd3^{-/-}$ LT-CML stem cells than in $Gdpd3^{+/+}$ LT-CML stem cells (Fig. 7a). Although the LRR-containing GPCR family member *Lgr5/Gpr49* is reportedly responsible for the maintenance of intestinal stem cells and cancer stem cells[32–34], the biological function of Lgr4/Gpr48 is not yet fully understood. These data therefore prompted us to investigate the role of Lgr4/Gpr48 in CML stem cells.

We transduced tet-CML-LSK cells with siRNAs targeting *Lgr4/Gpr48* mRNA and found that the colony-forming capacity of these cells was reduced (Fig. 7b; Supplementary Fig. 12). Because conventional $Lgr4/Gpr48^{-/-}$ mice are embryonic lethal[35], we used a mouse strain bearing a hypomorphic mutation of *Lgr4* established by gene-trap (Gt) methodology[36]. Levels of *Lgr4* mRNA in these mutants were decreased to 10% of levels in WT mice[36]. We then generated $Lgr4^{+/+}$ and $Lgr4^{Gt/Gt}$ tet-CML and retro-CML mice using the protocols described above. Absolute numbers of LT-CML stem cells were comparable between $Lgr4^{+/+}$ and $Lgr4^{Gt/Gt}$ tet-CML mice, whereas the frequency of these cells was modestly increased in $Lgr4^{Gt/Gt}$ tet-CML mice (Fig. 7c; Supplementary Fig. 13a). The colony-forming capacity of LT-CML stem cells isolated from $Lgr4^{Gt/Gt}$ tet-CML-affected mice decreased slightly compared to that of cells from $Lgr4^{+/+}$ tet-CML-affected controls (Fig. 7d). To evaluate the self-renewal capacity of these CML-initiating cells in vivo, we performed BMT of LSK cells from $Lgr4^{Gt/Gt}$ or $Lgr4^{+/+}$ retro-CML mice into irradiated recipients. Animals that received $Lgr4^{+/+}$ retro-CML-LSK cells developed CML disease just as shown in Fig. 2a. To our surprise, however, $Lgr4^{Gt/Gt}$ retro-CML-LSK cells displayed reduced disease-initiating capacity in transplanted recipients (Fig. 7e). At 20 days post-transplantation,

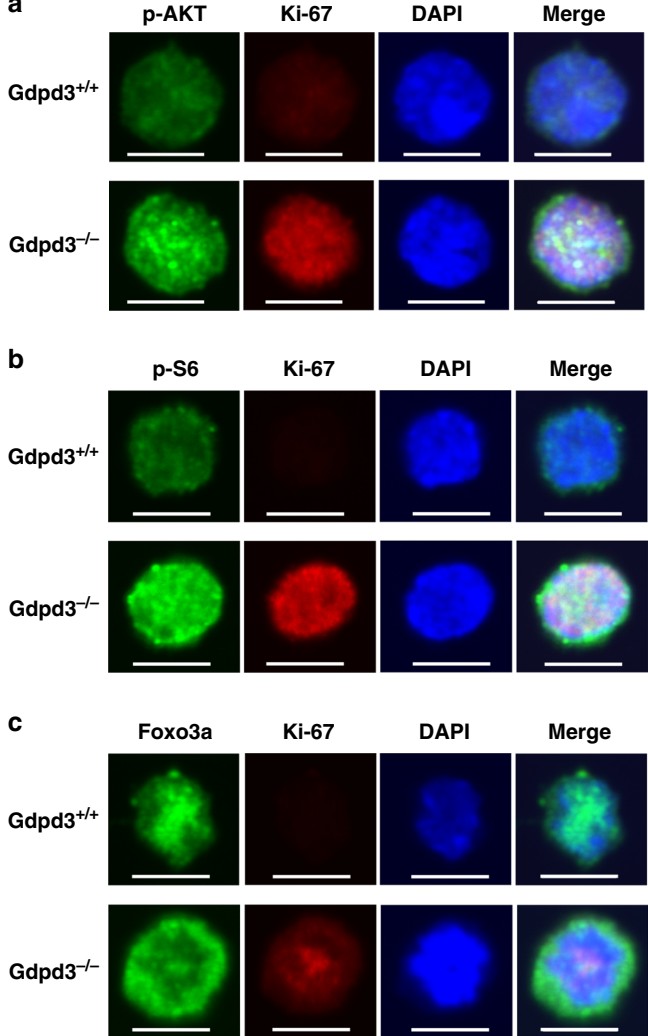

**Fig. 5 Gdpd3 deficiency activates the AKT/mTORC1 pathway and suppresses Foxo3a in LT-CML stem cells in vivo.** Phosphorylation of **a** AKT and **b** S6 ribosomal protein, and **c** nuclear localisation of Foxo3a in LT-CML stem cells that were isolated from *Gdpd3*[+/+] tet-CML-affected mice (four males), or *Gdpd3*[−/−] tet-CML-affected mice (three males), at 5 weeks post-Dox withdrawal. Cells were immunostained with mAbs to detect phosphorylated AKT (Ser473), phosphorylated S6 ribosomal protein (Ser235/236), Foxo3a, and Ki-67. Nuclei were visualised using DAPI. Scale bar, 10 μm. Results are representative of three biologically independent trials. (See also Supplementary Fig. 8).

the frequency and an absolute number of BCR-ABL1/EGFP[+] LSK cells in *Lgr4*[+/+] and *Lgr4*[Gt/Gt] retro-CML mice were comparable (Fig. 7f; Supplementary Fig. 13b). However, at 90 days post-transplantation, the frequency and an absolute number of BCR-ABL1/EGFP[+] LSK cells had decreased significantly in *Lgr4*[Gt/Gt] retro-CML mice compared with *Lgr4*[+/+] retro-CML mice (Fig. 7f; Supplementary Fig. 13b). In the same vein, the frequency of BCR-ABL1/EGFP[+] mature CML cells was markedly reduced in *Lgr4*[Gt/Gt] retro-CML mice compared with *Lgr4*[+/+] retro-CML mice (Fig. 7g). Thus, whereas Lgr4 is dispensable for the actual development of CML stem/progenitor cells, our results indicate that Lgr4 is important for their LT maintenance.

**Gdpd3 is essential for binding Foxo/β-catenin within nucleus.** Previous reports have demonstrated that Lgr4/Gpr48 functions as

a receptor for R-spondins in the Wnt/β-catenin signalling pathway[37–39], and that the *Ctnnb1* gene encoding β-catenin is crucial for the disease-relapsing capacity of CML stem cells[40]. We previously showed that Foxo3a is also essential for the self-renewal capacity of CML stem cells[25]. Lastly, Foxo3a and β-catenin are known to cooperatively regulate the metastasis of colon cancer cells[41]. This collection of facts prompted us to investigate if Gdpd3 or Lgr4, or both play a role in the nuclear interaction between Foxo3a and β-catenin in primitive LT-CML stem cells. We used the highly sensitive Duolink® in situ PLA technology to reveal binding between Foxo3a and active β-catenin within cell nuclei. As expected, we readily detected Foxo3a/β-catenin interaction in the nuclei of *Gdpd3*[+/+] LT-CML stem cells (Fig. 8a; Supplementary Fig. 14). However, no interaction between Foxo3a and active β-catenin was observed in the nuclei of LT-CML stem cells isolated from *Gdpd3*[−/−] tet-CML mice. Thus, Gdpd3 is involved in regulating the binding of Foxo3a to active β-catenin in LT-CML stem cells. Intriguingly, no Foxo3a/β-catenin binding was observed in LT-CML cells isolated from *Lgr4*[Gt/Gt] tet-CML mice, suggesting a potential mechanistic link.

Previous work has determined that β-catenin is activated in CML stem cells by PGE$_2$ via EP1[24]. We had already shown that PGE$_2$ was decreased in *Gdpd3*[−/−] CML BM cells compared to *Gdpd3*[+/+] CML BM cells (Fig. 4a). Thus, we focused on the effect of enforced PGE$_2$ treatment in vitro on the interaction between Foxo3a and β-catenin in LT-CML stem cells. PGE$_2$ treatment did not elevate Foxo3a/β-catenin binding in *Gdpd3*[+/+] LT-CML stem cells (Fig. 8b; Supplementary Fig. 15). Indeed, PGE$_2$ treatment only slightly increased this interaction in the nuclei of *Gdpd3*[−/−] LT-CML stem cells. In contrast, PGE$_2$ treatment dramatically increased Foxo3a/β-catenin interaction in the nuclei of *Lgr4*[Gt/Gt] tet-CML stem cells, exposing an interesting difference between *Gdpd3*[−/−] LT-CML stem cells and *Lgr4*[Gt/Gt] LT-CML stem cells that remains under investigation. In any case, our data collectively establish that Gdpd3 and lysophospholipid metabolism play multiple key roles in the maintenance of the stem cells driving CML.

## Discussion

Although signalling pathways related to lipid mediators are known to be involved in regulating CML stem survival[21–24], prior to our work, there had been little understood about how lipid biogenesis contributes to the self-renewal capacity of CML stem cells. In this study, we have shown that CML-LSK cells lacking the lysophospholipase D Gdpd3 display decreased self-renewal capacity in the second-round of serial BMT (Fig. 2b). More importantly, *Gdpd3*[−/−] CML-LSK cells show attenuated disease-relapsing capacity in animals treated with dasatinib (Fig. 6a). To our knowledge, this is the first demonstration that the *Gdpd3* gene involved in lysophospholipid metabolism is largely responsible for the maintenance of CML stem cells in vivo.

As illustrated in Fig. 8c, it is well known that Gdpd3 hydrolyses lysophospholipids back into LPAs[8,9]. Phospholipids are produced from LPAs via the Kennedy pathway (de novo pathway)[4] and are then converted into various lysophospholipids via the Lands' cycle (remodelling pathway)[5] (see also Fig. 1a). These mechanisms sustain the production of pleiotropic lysophospholipids and phospholipids, which contain many different combinations of fatty acid chains and polar bases. Lysophospholipase D activity appears to mainly contribute to lysophospholipid metabolism via LPA recycling. With respect to signalling downstream of Gdpd3-mediated lysophospholipid metabolism, we have shown that the AKT/mTORC1 pathway becomes more highly activated in *Gdpd3*[−/−] LT-CML stem cells compared with *Gdpd3*[+/+] LT-CML stem cells (Fig. 5a, b). In addition, *Gdpd3* deficiency triggers

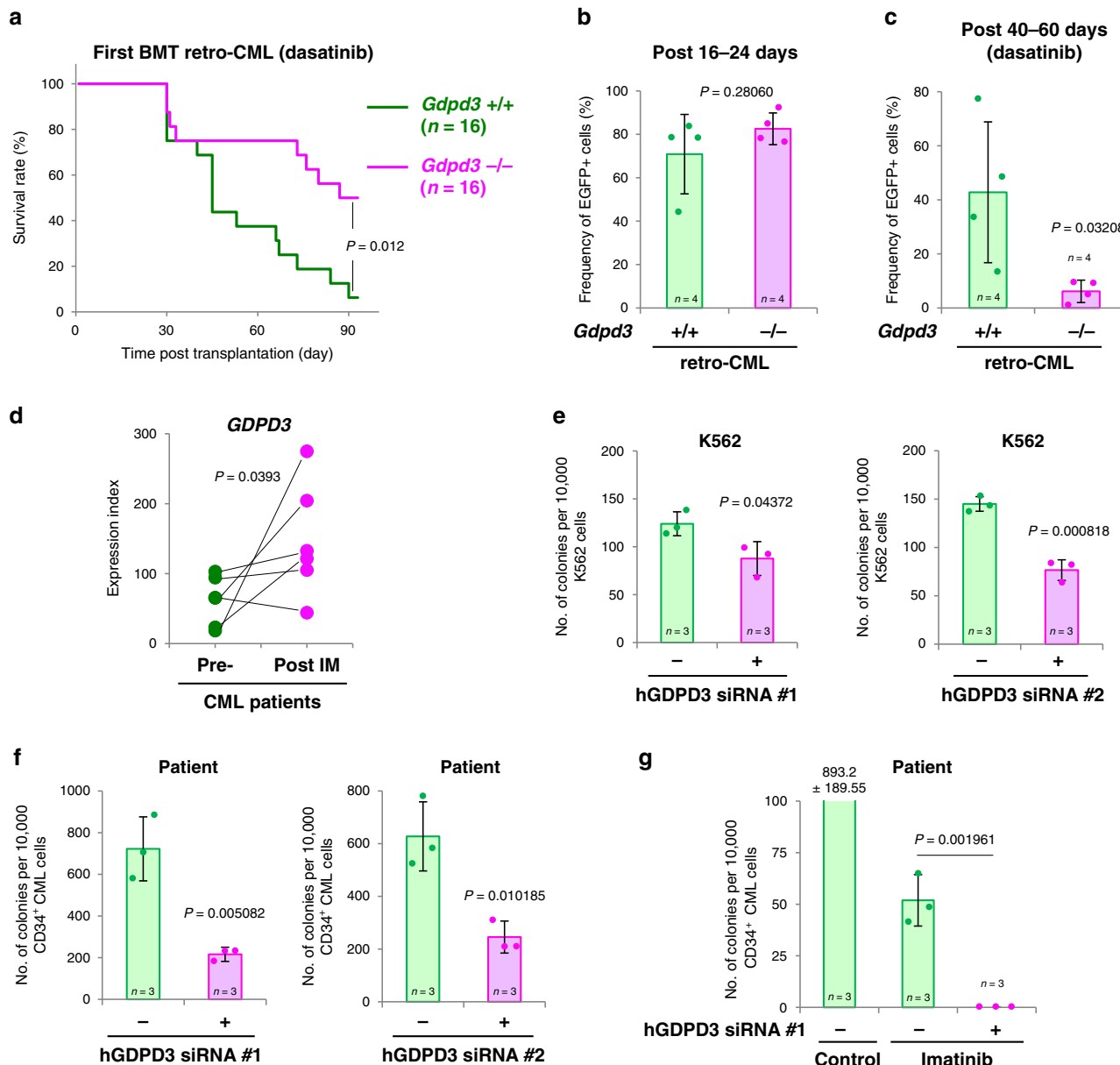

**Fig. 6 Gdpd3 is implicated in TKI-resistance in mouse and human CML stem cells. a** Survival rates of retro-CML-affected mice in a primary BMT experiment that received BCR-ABL1/EGFP-transduced LSK cells from either *Gdpd3*$^{+/+}$ mice (two males, two females) or *Gdpd3*$^{-/-}$ mice (three males, one female) as in Fig. 2a and were treated with dasatinib (5 mg kg$^{-1}$ per day) for days 8–60 post-transplantation. Results shown are cumulative data obtained from two independent experiments (*n* = 16 biologically independent recipient mice). Mouse survival was monitored for 90 days. (*P*-value, Log-rank non-parametric test). **b** Frequency of BCR-ABL1/EGFP$^+$leukaemia cells at 16–24 days post-transplantation in PB from retro-CML-affected mice that received BCR-ABL1/EGFP-transduced LSK cells from *Gdpd3*$^{+/+}$ mice (four females) or *Gdpd3*$^{-/-}$ mice (four females) as in Fig. 2a and were not treated with TKI (*n* = 4 biologically independent samples). Data are the mean frequency (%) ±s.d. of BCR-ABL1/EGFP$^+$leukaemia cells (*P*-value, unpaired two-sided Student's *t*-test). **c** Frequency of BCR-ABL1/EGFP$^+$leukaemia cells at 40–60 days post-transplantation in PB from retro-CML-affected mice that received BCR-ABL1/EGFP-transduced LSK cells from *Gdpd3*$^{+/+}$ mice (four females) or *Gdpd3*$^{-/-}$ mice (four females) as in Fig. 2a and were treated with dasatinib as in **a** (*n* = 4 biologically independent samples). Data are the mean frequency (%) ±s.d. of BCR-ABL1/EGFP$^+$leukaemia cells (*P*-value, unpaired two-sided Student's *t*-test). **d** Relative *GDPD3* mRNA expression in CD34$^+$ cells from CML patients pre- and post-imatinib (IM) therapy as determined by microarray analysis. Data are from a public database (GEO, GSE12211) (*n* = 6 biologically independent samples) (*P*-value, unpaired two-sided Student's *t*-test). **e**–**g** Quantitation of the colony-forming capacity of **e** K562 human CML cells and **f**, **g** primary human BM CD34$^+$ CML cells that were transduced with/without FITC-labelled siRNA targeting human *GDPD3* mRNA (hGDPD3 siRNA #1 or #2). FITC$^+$ and FITC$^-$ CML cells were purified at 3 days post-transduction and plated in semi-solid methylcellulose medium with/without imatinib (1 μM) for 14 days. Data are the mean colony number ± s.d. (*n* = 3). (*P*-value compared with control, unpaired two-sided Student's *t*-test). The relevant FACS data are shown in Supplementary Fig. 10. Results are representative of three biologically independent trials.

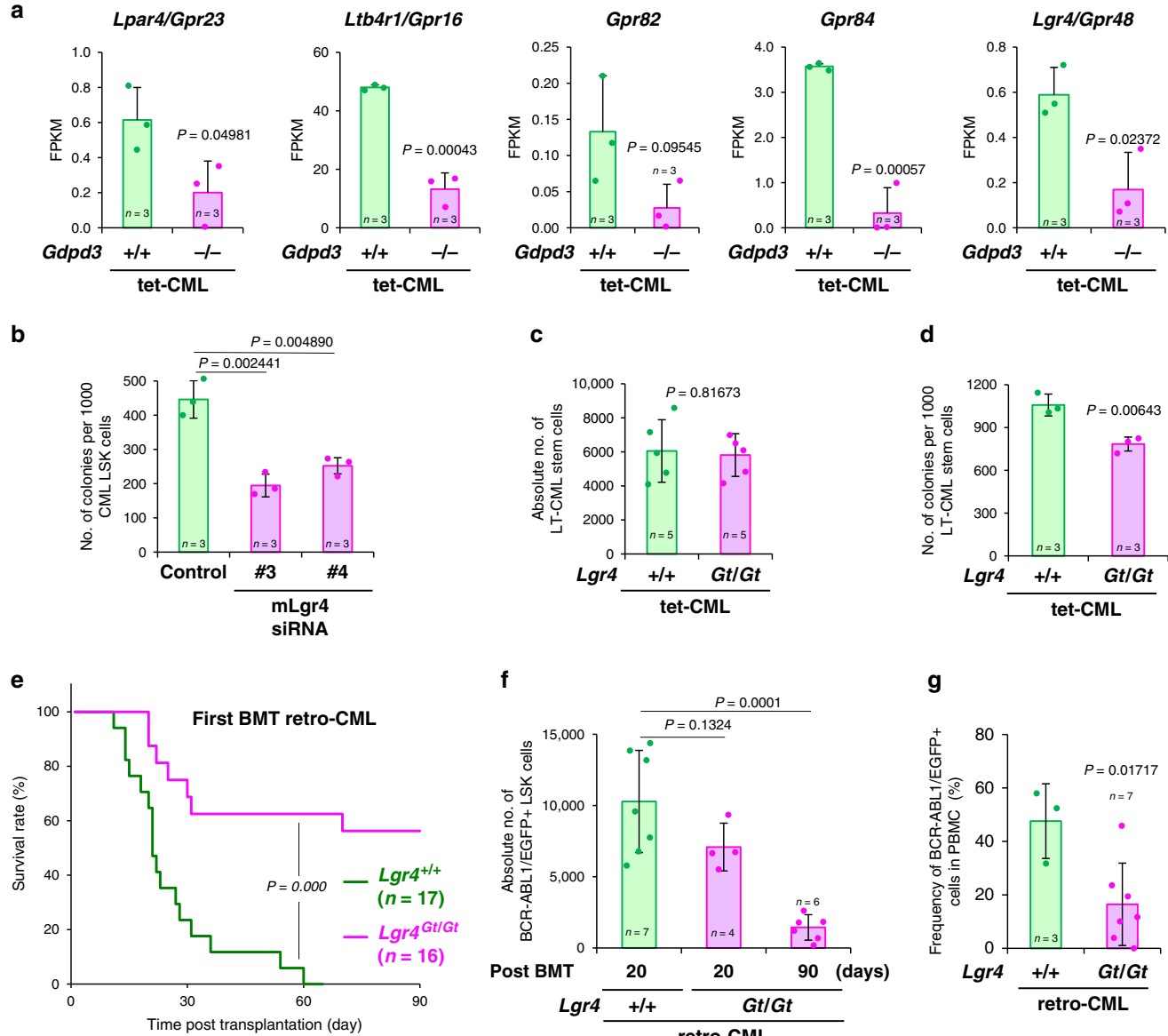

**Fig. 7 Lgr4/Gpr48 is involved in CML stem cell self-renewal in vivo. a** RNA-Seq determinations of mRNA levels of the indicated GPCR family genes in LT-CML stem cells isolated from *Gdpd3*[+/+] tet-CML-affected mice (22 males, 12 females) and *Gdpd3*[−/−] tet-CML-affected mice (five males, five females). Results are expressed as FPKM (see Methods). Data are the mean FPKM ± s.d. (*n* = 3 biologically independent samples) (*P*-value, unpaired two-sided Student's *t*-test). Results of the MA-plot and GO term enrichment analyses for these RNA-Seq data are shown in Supplementary Fig. 11a, b. **b** Quantitation of the colony-forming capacity of *Gdpd3*[+/+] CML-LSK cells that were transduced with/without Cy3-labelled siRNA targetting mouse *Lgr4/Gpr48* mRNA (mLgr4 #3 or Lgr4 #4). Cy3[+] and Cy3[−] CML-LSK cells were purified at 3 days post-transduction and plated in a semi-solid methylcellulose medium. Data are the mean colony number ± s.d. (*n* = 3) and are representative of three biologically independent experiments. (*P*-value compared with control, unpaired two-sided Student's *t*-test). The relevant FACS data are shown in Supplementary Fig. 12. **c** Absolute numbers of LT-CML stem cells isolated from BM of the two hind limbs of *Lgr4*[+/+] tet-CML-affected mice (four males, one female) and *Lgr4*[Gt/Gt] tet-CML-affected mice (five females) (*n* = 5 biologically independent samples). Data are the mean absolute numbers ± s.d. of LT-CML stem cells (*P*-value, unpaired two-sided Student's *t*-test). (See Supplementary Fig. 13a). **d** Quantitation of colony-forming capacity of LT-CML stem cells that were isolated from an *Lgr4*[+/+] tet-CML-affected mouse (one male) and an *Lgr4*[Gt/Gt] tet-CML-affected mouse (one male) and analysed as in Fig. 1f. Data are the mean colony number ± s.d. (*n* = 3) and are representative of three biologically independent experiments. (*P*-value, unpaired two-sided Student's *t*-test). **e** Survival rates of retro-CML-affected mice that received BCR-ABL1/EGFP-transduced LSK cells from *Lgr4*[+/+] mice (one male, one female) or *Lgr4*[Gt/Gt] mice (two females). Results shown are cumulative data from two independent experiments (*n* numbers indicate biologically independent recipient mouse numbers). Mouse survival was monitored for 90 days. (*P*-value, Log-rank non-parametric test). **f** Absolute numbers of BCR-ABL1/EGFP[+] CML-LSK cells isolated from BM of the two hind limbs of *Lgr4*[+/+] retro-CML-affected mice and *Lgr4*[Gt/Gt] retro-CML-affected mice after first-round BMT. Data are the mean absolute numbers ± s.d. of BCR-ABL1/EGFP[+] LSK cells (*n* numbers indicate biologically independent sample numbers) (*P*-value, unpaired two-sided Student's *t*-test). (See also Supplementary Fig. 13b). **g** Frequency of BCR-ABL1/EGFP[+] leukaemia cells post-BMT in PB of retro-CML-affected mice that received BCR-ABL1/EGFP-transduced LSK cells from *Lgr4*[+/+] mice (three females, *n* = 3 biologically independent samples) or *Lgr4*[Gt/Gt] mice (seven females, *n* = 7 biologically independent samples) as in Fig. 7e. Data are the mean frequency ± s.d. of BCR-ABL1/EGFP[+] cells (*P*-value, unpaired two-sided Student's *t*-test).

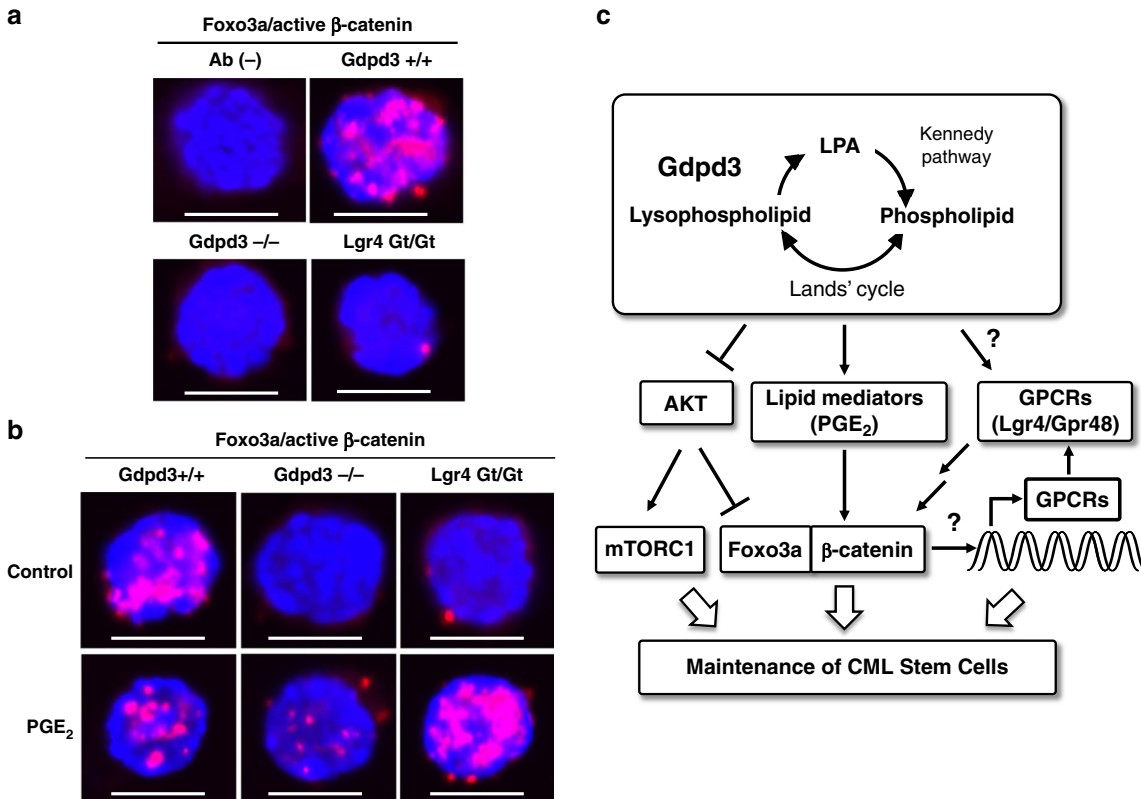

**Fig. 8 Gdpd3 and lysophospholipid metabolism are required for the maintenance of CML stem cells in vivo. a, b** Duolink® in situ PLA imaging to detect interaction between Foxo3a and active β-catenin in **a** freshly isolated LT-CML stem cells from *Gdpd3*+/+ tet-CML-affected mice (four males, two females), *Gdpd3*−/− tet-CML-affected mice (three males, two females), or *Lgr4*Gt/Gt tet-CML-affected mice (four females), at 5 weeks post-Dox withdrawal, and in **b** LT-CML stem cells that were isolated from *Gdpd3*+/+ tet-CML-affected mice (two females), *Gdpd3*−/− tet-CML-affected mice (two females), or *Lgr4*Gt/Gt tet-CML-affected mice (two females), at 5 weeks post-Dox withdrawal and treated in vitro with/without PGE$_2$ (10 μM) for 2 h. Results are representative of three biologically independent trials. Ab (-), technical negative control without primary antibody. Nuclei were visualised using DAPI. Scale bar, 10 μm. **c** Diagram outlining the proposed role of lysophospholipase D Gdpd3 in CML stem cells. Gdpd3 hydrolyses lysophospholipids to generate LPAs. *Gdpd3* deficiency in CML-affected mice results in decreased levels of certain LPAs and lipid mediators. Multiple downstream mechanisms, including (1) suppression of the AKT/mTORC1 pathway, (2) reduction of interaction between Foxo3a and β-catenin, and (3) altered expression of GPCR mRNAs such as Lgr4/Gpr48, may be involved in CML stem cell maintenance. Loss of *Gdpd3* is sufficient to decrease CML stem cell numbers in vivo, indicating that Gpdp3 plays a key role in maintaining the self-renewal capacity of CML stem cells in an oncogene-independent manner. Thus, Gdpd3-mediated lysophospholipid metabolism appears to be essential for supporting CML stemness.

cell division in CML stem cells and breaks the quiescence necessary to sustain CML stemness in vivo (Fig. 2g, h). Thus, our data indicate that Gdpd3 is a key suppressor of the AKT/mTORC1 pathway in CML stem cells, repressing their proliferation despite their expression of the *BCR-ABL1* oncogene. Our findings are consistent with reports stating that CML stem cells maintain their survival in an oncogene-independent manner[18], and support our hypothesis that Gdpd3-mediated lysophospholipid metabolism is crucial for this maintenance in vivo.

Our gene expression profiling studies showed that the expression of several GPCR genes was significantly decreased in *Gdpd3*−/− LT-CML stem cells compared to *Gdpd3*+/+ LT-CML stem cells. The GPCRs constitute the prototypical seven-transmembrane receptor family. GPCRs regulate diverse signal transduction pathways and so are attractive pharmacological targets. Drugs inhibiting some GPCRs have now been approved by the FDA[42–44]. Importantly, GPCR signalling has been implicated in the maintenance of embryonic stem (ES) cells, induced pluripotent stem (iPS) cells, somatic stem cells, and cancer stem cells[45,46]. More specifically, the GPCR Gpr84 is involved in regulating acute myelogenous leukaemia (AML) stem cells[47], and the GPCR Lgr5/Gpr49 contributes to the maintenance of intestinal stem cells and cancer stem cells[32–34]. Our data indicate that

Lgr4/Gpr48 is essential for the disease-initiating capacity of CML stem cells in vivo (Fig. 7e). At this time, the molecular mechanism by which Gdpd3 regulates the transcription of GPCR mRNAs remains under investigation. However, it is possible that lyso-phospholipid metabolism as shaped by Gdpd3 regulates Lgr4/Gpr48 mRNA levels in CML stem cells, which in turn might support the self-renewal capacity of these cells and thus their maintenance in vivo (Fig. 8c).

We found that several non-LPA lipid mediators, including PGE$_2$, eicosanoids, docosanoid, and Lyso-PAF, appeared to decrease in *Gdpd3*−/− CML-BM cells compared to *Gdpd3*+/+ CML-BM cells (Fig. 4), implicating Gdpd3-mediated lyso-spholipid metabolism in their production. Perhaps of relevance, our RNA-Seq analysis indicated that *Alox-15* expression is also decreased in *Gdpd3*−/− LT-CML stem cells compared to *Gdpd3*+/+ LT-CML stem cells (Supplementary Fig. 11c). The *Alox-15* gene encodes an arachidonate 15-lipoxygenase that is important for the maintenance of murine CML stem cells[23]. A reduction in levels of lipid mediators caused by *Gdpd3* deficiency might result in decreased *Alox-15* mRNA in CML stem cells, compromising their survival.

We were interested to note that enforced PGE$_2$ treatment in vitro rescued Foxo3a/β-catenin interaction in *Lgr4*Gt/Gt LT-CML stem

cells but not in $Gdpd3^{-/-}$ LT-CML stem cells. These divergent responses to $PGE_2$ are probably due to differing levels of intracellular lipid components, since $Gdpd3^{-/-}$ CML stem cells alter their production of several LPAs and lipid mediators, leading to the activation of the downstream AKT/mTORC1 pathway. In contrast, it is likely that $PGE_2$ treatment was able to restore the binding of Foxo3a to active β-catenin in $Lgr4^{Gt/Gt}$ CML stem cells because these cells suffer only from a decrease in Lgr4 mRNA expression. These data imply that, unlike Lgr4/Gpr48, Gdpd3 contributes to the maintenance of CML stemness via effects on multiple pathways, such as by suppressing AKT/mTORC1 signalling, decreasing the expression of GPCR genes, and regulating the interaction between Foxo3a and β-catenin (Fig. 8c).

In conclusion, we have demonstrated that the lysophospholipase D enzyme Gdpd3 is required for the maintenance of murine CML stem cells in vivo. We previously reported that TGFβ-FOXO signalling is also essential for CML stem cell maintenance[25]. Our future projects are now directed to investigate the molecular mechanisms by which Gdpd3-mediated lysophospholipid metabolism and TGFβ-FOXO signalling cooperate to support CML stemness in vivo.

## Methods

**CML mouse models.** A knockout mouse strain in which the $Gdpd3$ (NM_024228) gene was disrupted was generated by Setsurotech Inc. (Tokushima, Japan) using a genome-editing technique[48]. In brief, in vitro fertilised zygotes (C57BL/6 x C57BL/6) were electroporated with 100 ng μl$^{-1}$ recombinant Cas9 protein (Alt-R® S. p. Cas9 Nuclease, 1074181, Integrated DNA Technologies, Inc., Coralville, IA, USA), 100 ng μl$^{-1}$ crRNA (Alt-R® CRISPR-Cas9 crRNA, Integrated DNA Technologies), and 100 ng μl$^{-1}$ tracrRNA (Alt-R® CRISPR-Cas9 tracrRNA, Integrated DNA Technologies). The target sequence of Gdpd3 crRNA was (5′-GGCAGAACAAAGTAC AGGAG-AGG-3′) (Supplementary Fig. 3a). Electroporated zygotes were transferred into the oviduct of pseudopregnant female mice, and the mutant progeny were born on E19. Genotyping was carried out by hot-start PCR of genomic DNA from mouse tails. Two primer sets were used: Gdpd3 FWS3 5′-AGGCTATGATCCCTCTCCTG TACT-3′ and Gdpd3 R2 5′-CCTCTAGGCCCCAAGACTCT-3′ to detect the WT $Gdpd3$ allele, and Gdpd3 FB1 5′-GCTATGATCCCTCTCCTTTGTTCT-3′ and Gdpd3 R2 to detect the mutant (Δ5 bp) $Gdpd3$ allele. For PCR analysis, genomic DNA was subjected to 35 cycles of 30 s at 94 °C, 20 s at 64 °C, and 20 s at 72 °C to amplify the 298 bp product from the $Gdpd3$ WT allele, and was subjected to 35 cycles of 30 s at 94 °C, 20 s at 68 °C, and 20 s at 72 °C to amplify the 293 bp product from the $Gdpd3$ mutant allele (Supplementary Fig. 3c). Takara ExTaq® (Takara Bio Inc., Kusatsu, Shiga, Japan) was used for PCR. The $Lgr4/Gpr48$ gene-trap mouse strain was established as described (B6;CB-$Lgr4^{Gt(pUC-21)127Card}$, ID 1244, stocked at CARD, Kumamoto University, Kumamoto, Japan)[36]. The $Lgr4/Gpr48$ gene-trap mice were backcrossed for fourth generations in the C57BL/6 background. C57BL/6 mice were purchased from Crea Japan, Inc. (Tokyo, Japan).

To use a well-described tet-CML mouse model, $SCL$-$tTA$ transgenic mice (JAX database strain #006209) and $TRE$-$BCR$-$ABL1$ transgenic mice (JAX database strain #006202) were purchased from the Jackson Laboratory[27,28]. $SCL$-$tTA$ (C57BL/6; F5) and $TRE$-$BCR$-$ABL1$ (C57BL/6; F5) transgenic mice were interbred to generate $SCL$-$tTA$ x $TRE$-$BCR$-$ABL1$ double transgenic mice as described[26]. These animals were maintained in cages supplied with drinking water containing 20 mg L$^{-1}$ doxycycline (Dox; Sigma-Aldrich). At 5 weeks after birth, expression of the $BCR$-$ABL1$ oncogene was induced by replacing the Dox-containing drinking water with normal drinking water. The CML-like disease developed in the double transgenic mutants about 5 weeks after Dox withdrawal[26]. These animals were designated as tet-CML-affected mice in this study. To establish our $Gdpd3^{-/-}$ tet-CML-affected mouse model and our $Lgr4/Gpr48^{Gt/Gt}$ tet-CML mouse model, we crossed $Gdpd3^{-/-}$ mice (C57BL/6), or heterozygous $Lgr4/Gpr48^{Gt/Wt}$ mice (C57BL/6; F4), with $SCL$-$tTA$ x $TRE$-$BCR$-$ABL1$ double transgenic mice, respectively.

To establish our BCR-ABL1 transduction/transplantation-based CML model (retro-CML-affected mice), normal haematopoietic stem/progenitor (LSK) cells (4–5 × 10$^3$ cells per recipient mouse) isolated from WT (C57BL/6; CREA Japan, Inc.), $Gdpd3^{-/-}$, or $Lgr4/Gpr48^{Gt/Gt}$ mice were transduced with the human $BCR$-$ABL1$-$ires$ $EGFP$ retrovirus and transplanted into irradiated (9.5 Gy, Gammacell® 40 Exactor, Best Theratronics Ltd., Ottawa, Canada) recipient C57BL/6 mice along with BMMNCs (5 × 10$^5$ cells per recipient mouse) isolated from non-irradiated C57BL/6 mice. The CML-like disease developed in these recipients by 12–20 days post-transplantation[25].

To examine the in vivo effects of dasatinib administration, retro-CML-affected mice received vehicle alone [artificial gastric fluid solution (993 ml ddH$_2$O containing 2.0 g NaCl, 7 ml conc. HCl, and 3.2 g pepsin)], or dasatinib (Sprycel®; 5 mg kg$^{-1}$ day$^{-1}$; Bristol-Myers Squibb) in the vehicle. Treatment was delivered by oral gavage on days 8–60 post-transplantation. All animal care and experimentation were carried out in

accordance with the guidelines for animal and recombinant DNA experiments of Hiroshima University (Authorized Protocol Numbers A18-36, A18-37 and 30-257).

**Cell sorting.** BMMNCs were isolated from the two hind limbs of tet-CML-affected mice ($SCL$-$tTA^+$ $TRE$-$BCR$-$ABL1^+$) and WT healthy littermate mice ($SCL$-$tTA^+$) at 5 weeks after Dox withdrawal. To purify the most primitive LT-CML stem cells, BMMNCs were first blocked by incubation with anti-FcγIII/II receptor monoclonal antibody (mAb) (dilution 1:200, Clone # 2.4G2, Cat. # BD 553142, Lot # 79813, BD Biosciences), and then stained with anti-CD4-FITC (dilution 1:200, Clone # RM4-5, Cat. # 11-0042-86, Lot #E00084-1631, eBioscience), anti-CD8a-FITC (dilution 1:200, Clone # 53-6.7, Cat. # 11-0081-86, Lot # E00117-1632, eBioscience), anti-B220-FITC (dilution 1:200, Clone # RA3-6B2, Cat. # 11-0452-86, Lot #E00310-1631, eBioscience), anti-Mac1-FITC (dilution 1:200, Clone # M1/70, Cat. #11-5931-86, Lot # E00740-1631, eBioscience), anti-Gr-1-FITC (dilution 1:200, Clone # RB6-8C5, Cat. # 11-0112-86, Lot #E00150-1632, eBioscience), anti-TER119-FITC (dilution 1:200, Clone # Ly-76, Cat. #11-5921-85, Lot #E00736-1630, eBioscience), anti-Sca-1-PE (dilution 1:200, Clone # E13-161.7, Cat. # BD553336, Lot # 05152, BD Biosciences), anti-cKit-APC (dilution 1:200, Clone # ACK2, Cat. # 17-1172-83, Lot # E17176-101, eBioscience), anti-CD135/Flk2/Flt3-biotin (dilution 1:200, Clone # A2F10, Cat. # 13-1351-85, Lot # E02732-1630, eBioscience), anti-CD48-APC-Cy7 (dilution 1:250, Clone # HM48-1, Cat. # 103432, Lot # B173122, BioLegend), and anti-CD150/SLAM-Pacific blue (dilution 1:250, Clone # TC15-12F12.2, Cat. # 115924, Lot # B224451, BioLegend) mAbs. Biotinylated primary mAbs were visualised using Streptavidin-PE-Cy7 (dilution 1:400, Cat. # 557598, Lot # 6112577, BD Biosciences). CD150$^+$CD48$^-$CD135$^-$LSK (Lineage$^-$Sca-1$^+$cKit$^+$) cells were purified using a FACS Aria III cell sorter (S/N, P64828201002) with BD FACSDiva software ver. 6.1.3 (BD Biosciences).

To purify WT haematopoietic stem/progenitor (LSK) cells, BMMNCs isolated from WT, $Gdpd3^{-/-}$, and $Lgr4^{Gt/Gt}$ mice (6–8-wk old) were first blocked by incubation with anti-FcγIII/II receptor mAb (dilution 1:200, Clone # 2.4G2, Cat. # BD 553142, Lot # 79813, BD Biosciences), and then stained with anti-CD4-FITC (dilution 1:200, Clone # RM4-5, Cat. # 11-0042-86, Lot #E00084-1631, eBioscience), anti-CD8a-FITC (dilution 1:200, Clone # 53-6.7, Cat. # 11-0081-86, Lot # E00117-1632, eBioscience), anti-B220-FITC (dilution 1:200, Clone # RA3-6B2, Cat. # 11-0452-86, Lot #E00310-1631, eBioscience), anti-Mac1-FITC (dilution 1:200, Clone # M1/70, Cat. #11-5931-86, Lot # E00740-1631, eBioscience), anti-Gr-1-FITC (dilution 1:200, Clone # RB6-8C5, Cat. #11-0112-86, Lot #E00150-1632, eBioscience), anti-TER119-FITC (dilution 1:200, Clone # Ly-76, Cat. #11-5921-85, Lot #E00736-1630, eBioscience), anti-Sca-1-PE (dilution 1:200, Clone # E13-161.7, Cat. # BD553336, Lot # 05152, BD Biosciences), and anti-cKit-APC (dilution 1:200, Clone # ACK2, Cat. # 17-1172-83, Lot # E17176-101, eBioscience). WT LSK cells were purified using a FACS Aria III cell sorter (BD Biosciences).

For serial transplantation of CML-LSK cells (see below), MNCs isolated from the two hind limbs and spleen of retro-CML-affected mice were first blocked by incubation with anti-FcγIII/II receptor mAb (dilution 1:200, Clone # 2.4G2, Cat. # BD 553142, Lot # 79813, BD Biosciences), and then stained with anti-CD4-biotin (dilution 1:200, Clone # RM4-5, Cat. # 13-0042-86, Lot #E02364-369, eBioscience), anti-CD8a-biotin (dilution 1:200, Clone # 53-6.7, Cat. # 13-0081-86, Lot #E02387-339, eBioscience), anti-B220-biotin (dilution 1:200, Clone # RA3-6B2, Cat. # 13-0452-86, Lot #E02532-301, eBioscience), anti-Mac1-biotin (dilution 1:200, Clone # M1/70, Cat. #13-0112-86, Lot #E033770, eBioscience), anti-Gr-1-biotin (dilution 1:200, Clone # RB6-8C5, Cat. #13-5931-86, Lot # E033865, eBioscience), anti-TER119-biotin (dilution 1:200, Clone # Ly-76, Cat. #13-5921-85, Lot # 4300555, eBioscience), anti-Sca-1-PE (dilution 1:200, Clone # E13-161.7, Cat. # BD553336, Lot # 05152, BD Biosciences), and anti-cKit-APC (dilution 1:200, Clone # ACK2, Cat. # 17-1172-83, Lot # E17176-101, eBioscience) mAbs. Biotinylated primary mAbs were visualised using Streptavidin-PE-Cy7 (dilution 1:400, Cat. # 557598, Lot # 6112577, BD Biosciences). BCR-ABL1/EGFP$^+$ CML-LSK cells were purified using a FACS Aria III cell sorter (BD Biosciences).

For siRNA transductions (see below), CML-LSK cells were purified from BMMNCs of tet-CML-affected mice using a FACS Aria III cell sorter (BD Biosciences).

**Serial transplantation of CML stem cells.** To evaluate the retention of disease-initiating capacity by CML-initiating cells in vivo, secondary transplantation of BCR-ABL1/EGFP$^+$ CML-LSK cells was performed. BCR-ABL1/EGFP$^+$ CML-LSK cells (3 × 10$^4$) freshly purified from primary BMT recipients were serially transplanted into a second set of lethally irradiated (9.5 Gy) congenic recipient mice along with 5 × 10$^5$ WT BMMNCs from C57BL/6 mice[25]. Mouse survival and disease relapse were monitored for up to 90 days.

**Flow cytometry and blood cell count.** MNCs isolated from PB of $Gdpd3^{+/+}$ retro-CML-affected mice, $Gdpd3^{-/-}$ retro-CML-affected mice as well as from a normal healthy C57BL/6 mouse (negative control), were stained with anti-CD4-PE-Cy7 (dilution 1:500, Clone # RM4-5, Cat. # 25-0042-82, Lot #E07503-1630, eBioscience), anti-CD8a-PE-Cy7 (dilution 1:500, Clone # 53-6.7, Cat. # 25-0081-82, Lot # E07510-1631, eBioscience), anti-B220-PE (dilution 1:500, Clone # RA3-6B2, Cat. # 553089, Lot # 76923, BD Biosciences), anti-Mac1-APC (dilution 1:500, Clone # M1/70, Cat. # 17-0112-82, Lot #E07073-1631, eBioscience), and

anti-Gr-1-APC (dilution 1:500, Clone # RB6-8C5, Cat. # 17-5931-82, Lot # E07334-1630, eBioscience) mAbs. BCR-ABL1/EGFP$^+$ cells were evaluated using a FACS Aria III instrument and the t-SNE algorithm[49] in FlowJo$^{TM}$ (build number 10.6.1) software. BCR-ABL1/EGFP$^+$ cells in PB isolated from a $Gdpd3^{+/+}$, $Gdpd3^{-/-}$, $Lgr4^{+/+}$, and $Lgr4^{Gt/Gt}$ retro-CML-affected mouse were evaluated using a FACS Aria III instrument. The absolute number of CML-LSK cells in the spleen and BM of a retro-CML-affected mouse was calculated as [total number of MNCs isolated from either the BM of the two hind limbs or the spleen × frequency of BCR-ABL1/EGFP$^+$ CML-LSK cells (%) × 1/100]. The absolute number of LT-CML stem cells in BM of a tet-CML mouse was calculated as [total number of MNCs isolated from BM of two hind limbs × frequency of CD150$^+$CD48$^-$CD135$^-$LSK cells (%) × 1/100]. For analysis of blood cell counts. PB from the postorbital vein was analysed by a particle counter PCE-310 (ERMA Inc., Tokyo Japan).

**Lipidomics.** For lipidomics analyses, total BMMNCs ($1 \times 10^7$) were isolated from $Gdpd3^{+/+}$ tet-CML-affected mice (five females), $Gdpd3^{-/-}$ tet-CML-affected mice (one male and three females) at 5-week post-Dox withdrawal as described above. Total BMMNCs ($1 \times 10^7$) were also isolated from 6–8-wk old WT C57BL/6 mice (two males and four females). CML-LSK cells ($5 \times 10^5$) were isolated from $Gdpd3^{+/+}$ tet-CML-affected mice (four males and 11 females), or from $Gdpd3^{-/-}$ tet-CML-affected mice (six males and ten females), at 5 weeks post-Dox withdrawal as described above. Cell pellets were frozen at −80 °C immediately after centrifugation. Lipidomics analyses were performed by Shimadzu Techno-research Inc. (Kyoto, Japan). LPAs were measured using a NexeraX2 system (Shimadzu Corporation, Kyoto, Japan) and Triple Quad 5500 (Sciex, Framingham, MA, USA) (Source Data file, Supplementary Method 1). Lipid mediators were measured using a NexeraX2 System and an LCMS-8050 liquid chromatograph-mass spectrometer system (Shimadzu)[50] (Source Data file, Supplementary Method 2, 3). Data were compiled using the Lipidomediator LC/MS/MS Method Package (Ver. 3) (Shimadzu Corporation). Lipidomics data analyses were performed using Traverse MS Ver. 1.2.7. (Reifycs Inc., Tokyo, Japan).

**RNA sequencing and bioinformatics.** LT CML stem cells isolated from $Gdpd3^{+/+}$ tet-CML-affected mice and $Gdpd3^{-/-}$ tet-CML-affected mice were directly sorted into 500 μl Isogene solution (Nippon Gene, Co. Ltd., Toyama, Japan). RNA extraction and sequencing were performed by DNAFORM (Yokohama, Japan). The RNA quality was confirmed using an Agilent 2100 Bioanalyzer (Agilent Technologies). All RNA samples had an RNA integrity number of >7.5 and exceeded the quality threshold for RNA sequencing. Libraries were constructed from total RNA using the SMART-Seq® v4 Ultra Low Input RNA Kit for Sequencing (Takara Clontech, Kusatsu, Shiga, Japan). Full-length double-stranded cDNA libraries were fragmented by transposase using the Nextera XT DNA Library Preparation Kit (Illumina, San Diego, CA, USA). Paired-end reads of 150 bases were generated using HiSeq X Ten (Illumina). Sequence reads in Fastq format were assessed for quality using FastQC. Filtered reads were mapped to the reference genome related to the species using STAR v.2.6.1a alignment software[51]. Gene expression levels were measured with the Bioconductor package DESeq2 (ver. 1.20.0)[52] (https://bioconductor.org/packages/release/bioc/html/DES0eq2.html) using the Ensembl database (https://ensembl.org/index.html) and quantified as the ratio of reads mapped to a gene to the gene length in kbp and expressed as the fragments per kb of transcript per million fragments mapped (FPKM). The Bioconductor package DESeq2 (ver. 1.20.0) was also used for calculating BaseMean, FoldChange, $P$-value (two-tailed Wald test)[52], and adjusted $P$-value (Padj, two-tailed Wald test)[52,53]. MA-plots were created using the Bokeh library (ver. 0.13.0) (https://docs.bokeh.org/en/0.13.0/). GO enrichment analyses were performed using the DAVID Bioinformatics Resource 6.8. (http://david.abcc.ncifcrf.gov). The RNA-sequencing data of $Gdpd3^{+/+}$ and $Gdpd3^{-/-}$ LT-CML stem cells are available from a public database GEO (GEO, ID: GSE149442) in NCBI, NIH, USA (https://www.ncbi.nlm.nih.gov/gds/).

**Quantitative real-time RT-PCR analysis.** The RNeasy kit (QIAGEN, Venlo, Netherlands) was used to purify RNA samples from LT-stem cells ($4–5 \times 10^4$) isolated from six $Gdpd3^{+/+}$ tet-CML-affected ($SCL$-$tTA^+TRE$-$BCR$-$ABL1^+$) mice and eight healthy littermate control ($SCL$-$tTA^+$) mice at 5 weeks post-Dox withdrawal. RNA samples were reverse-transcribed using the Advantage RT-for-PCR kit (Takara-Clontech). Real-time quantitative PCR was performed using SYBR green *Premix EX Taq$^{TM}$* (Takara) on an Mx3000P® Real-time PCR system (Stratagene). The following primers were used: 5′-CAT TTC GCT GTG GAT GGA TGA-3′, and 5′-CCA TTG GCT CCC AAG CTG A-3″ for $Gdpd3$; 5′-AGG TCA TCA CTA TTG GCA ACG A-3′ and 5′-CAC TTC ATG ATG GAA TTG AAT GTA GTT-3′ for $Actb$ (β-actin). The following cycle parameters were used: denaturation at 95 °C for 10 s, and annealing and elongation at 60 °C for 30 s for $Gdpd3$, and 57 °C for 30 s for $Actb$.

**Colony-forming assays.** OP-9 murine stromal cells were purchased from ATCC® (Lot # 70019055, CRL-2749). LT-CML stem cells ($1 \times 10^3$ per plate) were co-cultured on OP-9 stromal cells in 3% $O_2$ at 37 °C for 5 days[25]. Cells were harvested, washed in PBS, and plated in semi-solid methylcellulose medium containing SCF, IL-3, IL-6 and erythropoietin (Methocult GF M3434; Stem Cell Technologies). WT

haematopoietic stem/progenitor (LSK) cells were cultured in semi-solid methylcellulose medium (Methocult GF M3434). After growth for 7 days under hypoxic (3% $O_2$) conditions at 37 °C, colony numbers were counted under a light microscope.

**Competitive reconstitution assay.** C57BL/6-CD45.1 mouse strain was purchased from Sankyo Labo Service Corporation, Inc. (Tokyo, Japan). Lethally irradiated C57BL/6-CD45.1 congenic recipient mice were transplanted with $2 \times 10^3$ normal haematopoietic LSK cells isolated from normal healthy $Gdpd3^{+/+}$ and $Gdpd3^{-/-}$ mice (C57BL/6-CD45.2) plus $5 \times 10^5$ unfractionated BMMNCs from healthy C57BL/6-CD45.1 mice. Reconstitution of donor-derived cells (CD45.2) was monitored by staining blood cells with mAbs against FITC-CD45.2 (dilution 1:200, Clone # 104, Cat. # 11-0454-85, Lot # E00316-130, eBioscience), and PE-CD45.1 (dilution 1:200, Clone # A20, Cat. # BD553776, Lot # 17194, BD Biosciences) using a FACS Aria III instrument. For serial transplantation analyses, $2 \times 10^3$ normal LSK cells were obtained from recipient mice at 16 weeks post-transplantation (first-BMT) and transplanted into a second set of lethally irradiated C57BL/6-CD45.1 congenic mice (second-BMT) using the same BMT protocol.

**GDPD3 mRNA expression in human CML patients.** Data on $GDPD3$ mRNA levels in human CML patients were obtained from a public database GEO (GEO, ID: GSE12211)[30] that contains microarray analyses of the CD34$^+$ cells in PB from six CML patients before treatment, and the same six CML patients after 7 days of treatment with IM (400 mg kg$^{-1}$ day$^{-1}$).

**RNA interference in mouse CML-LSK cells.** The following Cy3-labelled siRNA duplexes targeting $Gdpd3$ mRNA were synthesised by Dharmacon Inc. (Lafayette, CO, USA): 5′-Cy3-GAG CAG AUC UCU UGG AAU UUU-3′ and 5′-AAU UCC AAG AGA UCU GCU CUU-3′ for mouse $Gdpd3$ siRNA #1; 5′-Cy3-GGU CAA AGA AAG AAA UGA AUU-3′ and 5′-UUC AUU UCU UUC UUU GAC CUU-3′ for mouse $Gdpd3$ siRNA #3. The following Cy3-labelled siRNA duplexes targeting $Lgr4/Gpr48$ mRNA were synthesised by Dharmacon Inc.: 5′-Cy3-GAA CAU AGC CAA AUA AUC AUU-3′ and 5′-UGA UUA UUU GGC UAU GUU CUU-3′ for mouse $Lgr4$ siRNA #3; 5′-Cy3-AAA GAA GAC CCG UCA GAA AUU-3′ and 5′-UUU CUG ACG GGU CUU CUU UUU-3′ for mouse $Lgr4$ siRNA #4. $Gdpd3^{+/+}$ tet-CML-LSK cells were transfected for 3 h under hypoxic (3% $O_2$) conditions with Cy3-labelled siRNA duplexes using Lipofectamine® RNAiMAX (Thermo Fisher Scientific) in serum-free stem cell medium SF-03 that lacked cytokines and penicillin/streptomycin. Transfected cells were cultured in serum-free stem cell medium SF-03 containing BSA, TPO (100 ng ml$^{-1}$), SCF (100 ng ml$^{-1}$), and penicillin/streptomycin for 2 days under hypoxic (3% $O_2$) conditions. Cy3-negative (control) and Cy3-positive cells were sorted using a FACS Aria III instrument. To determine colony-forming capacity, Cy3-negative and Cy3-positive CML-LSK cells were cultured in semi-solid methylcellulose medium (Methocult GF M3434; Stem Cell Technologies). Colonies were counted 7 days later as above.

**RNA interference in human CML CD34$^+$ cells.** The following FITC-labelled siRNA duplexes targeting human $GDPD3$ mRNA were synthesised by Dharmacon Inc.: 5′-FITC- AGG AGA AGC UGG AGG UUU AUU-3′ and 5′-UAA ACC UCC AGC UUC UCC UUU-3′ for human $GDPD3$ #1; 5′-FITC-CCA UGA GCG UAG AGA UCA AUU-3′ and 5′-UUG AUC UCU ACG CUC AUG GUU-3′ for human $GDPD3$ siRNA #2. Viable BMMNCs from a chronic phase CML patient were obtained at the University Hospital, Dokkyo Medical University (IRB approval number: 26058). Primary CML cells were stained with mouse anti-CD34-APC mAb (dilution 1:100, Clone # 8G12, Cat. # 340441, Lot # 6183704, BD Biosciences) and sorted using a FACS Aria III cell sorter (BD Biosciences). CD34$^+$ CML cells were transfected for 3 h under hypoxic (3% $O_2$) conditions with FITC-labelled siRNA duplexes using Lipofectamine® RNAiMAX (Thermo Fisher Scientific) in serum-free medium OPTI-MEM®I (Gibco) that lacked cytokines and penicillin/streptomycin. Transfected cells were cultured in serum-free stem cell medium SF-03 containing BSA, TPO (100 ng ml$^{-1}$), SCF (100 ng ml$^{-1}$), and penicillin/streptomycin for 3 days under hypoxic (3% $O_2$) conditions. FITC-negative (control) and FITC-positive cells were sorted using a FACS Aria III instrument. K562 CML cells were purchased from ATCC® (Lot # 70016362, CCL-243) and transfected with FITC-labelled siRNA duplexes as above. The colony-forming abilities of transfected primitive human CML CD34$^+$ cells and K562 cells were evaluated by culture in semi-solid methylcellulose medium containing SCF, GM-CSF, IL-3, IL-6, G-CSF and erythropoietin (Methocult GF$^+$ H4435; Stem Cell Technologies). After growth for 14 days at 37 °C under hypoxic (3% $O_2$) conditions without or with 1 μm imatinib (Axon Medchem, Groningen, Netherlands), colony numbers were counted under a light microscope.

**Cell cycle analysis.** To determine the cell cycle status of CML-LSK cells in vivo, retro-CML-affected mice were intraperitoneally administered BrdU (100 mg kg$^{-1}$ of body weight in saline; Sigma) for 3 h. CML-LSK cells were recovered as described above and stained with mouse anti-BrdU-FITC mAb (dilution 1:500, Clone # 3D4, Cat. # 51-23614 L, Lot # 7222635, BD Biosciences) plus 7AAD (BD Biosciences) by the standard procedure of FITC BrdU Flow Kit (559619, BD

Pharmingen™). The cell cycle distribution of CML-LSK cells was determined using a FACS Aria III instrument.

**Fluorescence immunostaining**. For immunostaining, LT-CML stem cells that were freshly isolated from tet-CML-affected mice were immediately fixed with 4% paraformaldehyde for one hour. Fixed cells were permeabilised with 0.25% Triton-X100 for 15 min, washed, and blocked by incubation in 2% BSA in TBS for one hour. Blocked cells were incubated overnight at 4 °C with rabbit anti-phospho-AKT (Ser473) (dilution 1:50, Clone # D9E, Cat. # 4060S, Lot# 5, Cell Signalling Technology, Danvers, MA, USA), rabbit anti-phospho-S6 ribosomal protein (Ser235/236) (dilution 1:50, Clone # D57.2.2E, Cat. # 4858S, Lot# 11, Cell Signalling Technology), rabbit anti-Foxo3a (dilution 1:25, Clone 75D8, Cat. #2497S, Lot # 2, Cell Signalling Technology) and mouse (dilution 1:50, Clone # B56, Cat. # 550609, Lot# 43365, BD Pharmingen) mAbs. Primary mAbs were visualised by incubating the cells with AlexaFluor 546-conjugated goat anti-mouse IgG (dilution 1:200, Cat. # A11030, Lot # 833292, Molecular Probes®, Life technologies, Eugene, OR, USA), or AlexaFluor 647-conjugated goat anti-rabbit IgG (dilution 1:200, Cat. # A21245, Lot # 927083, Molecular Probes®) antibodies (Abs). Nuclei were stained with the DNA marker DAPI (Sigma). Stained slides were mounted using Prolong Diamond® (Thermo Fisher Scientific, Waltham, MA, USA), and fluorescent images were acquired using confocal microscopy (FV10i, Olympus Corporation, Tokyo, Japan) and Photoshop software (CS4 Ver11.0.2. Adobe).

**Duolink® in situ proximity ligation assay**. To examine interactions between Foxo3a and active β-catenin, we used the Duolink® in situ PLA system (Merck, Kenilworth, NJ, USA)[26]. LT-CML stem cells that were freshly isolated from tet-CML-affected mice were immediately fixed with 4% paraformaldehyde for 60 min. For PGE₂ treatment, LT-CML stem cells were incubated in 3% O₂ at 37 °C for 2 h with the appropriate vehicle control or 10 μM PGE₂ (Item: 14010, Batch:0533515-113, Cayman Chemical Company, Ann Arbor, MI, USA). Treated cells were fixed with 4% paraformaldehyde for 60 min. Fixed cells were permeabilised with 0.25% Triton-X100 for 15 min, washed, and blocked by incubation in 2% BSA in TBS for one hour. Blocked cells were incubated overnight at 4 °C with rabbit anti-Foxo3a (dilution 1:25, Clone 75D8, Cat. #2497S, Lot # 2, Cell Signalling Technology) and mouse anti-active β-catenin (dilution 1:25, Clone 8E7, Cat. # 05-665, Lot #, 2700799 and 3270747, Millipore, Temecula, CA, USA) mAbs. The proximate binding of these Abs was then detected using PLA secondary Abs. Stained slides were mounted using Duolink® in situ Mounting Medium with DAPI (Merck), and fluorescent images were acquired by confocal microscopy (FV10i, Olympus) and Photoshop software (CS4 Ver11.0.2. Adobe).

**Statistical analyses and reproducibility**. For RNA-sequencing in Fig. 7a and Supplementary Figs. 11a–c, FPKM, BaseMean, FoldChange, P-value (two-tailed Wald test)[52], and adjusted P-value (two-tailed Wald test)[52,53] were determined by the Bioconductor package DESeq2 (ver. 1.20.0) (https://bioconductor.org/packages/release/bioc/html/DES0eq2.html)[52]. For the Kaplan-Meier analyses for the survival of CML-affected mice in Figs. 1g, 2a, b, 6a, 7e, statistical differences were determined using the log-rank non-parametric test (IBM SPSS Statics 23, IBM, Chicago, IL, USA). For Fig. 3a, b, 4a–c, the unpaired one-sided Student's t test was used (Microsoft® Excel Ver. 16.40, Redmond, WA, USA). For the other comparisons, the unpaired two-sided Student's t test was used (Microsoft® Excel Ver. 16.40). Data were indicated as the means ± standard deviation (s.d.). P < 0.05 was considered statistically significant. The lipidomics analyses and RNA-sequencing were based on every single experiment. For Figs. 1d, 2a, 6a, 7e, these experiments were from two independent experiments. The other experiments were repeated at least three times. No statistical method was used to predetermine the sample size. The sample size was determined on the basis of literature in this research field[22–28,40].

**Note added in proof**. During the reviewing process, a paper demonstrating that Lgr4 is essential for self-renewal in acute myelogenous leukaemia (AML) stem cells was published[54]. The study supports our conclusion about the role of Lgr4 in maintenance of CML stem cells.

**Reporting summary**. Further information on research design is available in the Nature Research Reporting Summary linked to this article.

## Data availability

For lipidomics analyses data in Figs. 3a, b and 4a–c, the original data are available from Source Data file, and Supplementary Method 1–3. For RNA-sequencing data in Fig. 7a, and Supplementary Fig. 1, 11a–c, our data are available from a public database gene expression omnibus (GEO, ID: GSE70031 and GSE149442, NCBI, NIH, USA) and Source Data file. Gene expression levels were measured with DESeq2 (ver.1.20.0) (https://bioconductor.org/packages/release/bioc/html/DESeq2.html)[52] using the Ensembl database (https://ensembl.org/index.html). MA-plots were created using the Bokeh library (ver. 0.13.0) (https://docs.bokeh.org/en/0.13.0/). GO enrichment analyses were performed using the DAVID Bioinformatics Resource 6.8. (http://david.abcc.ncifcrf.gov).

Gene expression data for Fig. 6d was downloaded from the public microarray dataset in the GEO database under accession code GSE12211[30]. The source data for Figs. 1b–g, 2a–g, 3a, b, 4a–c, 6a, 7a–g, Supplementary Figs. 1, 3b, c, 4a–c, 5a, 6a–d, 7, 11a–c, 13a, b, and Supplementary Table 1 have been provided as Source Data file. Source data are provided with this paper.

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

## Acknowledgements

We thank D.G. Tenen for the *SCL-tTA* and *TRE-BCR-ABL1* transgenic mouse strains, H. Nakauchi for the C57BL/6-CD45.1 mouse strain, H. Matsushima and Y. Yamashita for genome-editing, K. Watanabe for lipidomics, Y. Takegami and N. Tominaga for RNA-Seq, M. Iwamoto for *Lgr4^Gt/Gt* mouse strain, H. Honda for BCR-ABL1 cDNA, and T. Kitamura for Plat-E retroviral packaging cells. K.N. was supported by a Grant-in-Aid for Scientific Research (B) (KAKENHI Grant Numbers 17H0357800 and 20H0351700) from MEXT, the Government of Japan, and a grant from the Princess Takamatsu Cancer Research Fund (Grant Number 17-24920). S.-J.K. was supported by a grant from the Korea Health Technology R&D Project through the National Cancer Center (HA17C0037), the Ministry of Health and Welfare, Republic of Korea.

## Author contributions

K.N. designed and performed experiments, analysed data, and co-wrote the paper. R.O., E.M. and C.K. performed lipidomics analyses. K.-M.Y. performed bioinformatic analyses. T.H., M.A. and K.A. provided the *Lgr4^Gt/Gt* mouse strain. K.M. and K.S. provided viable BMMNCs from a human CML patient. Y.S. supported animal experiments. D.-W.K. supported CML research. A.O. and S.-J.K. designed experiments and co-wrote the paper.

## Competing interests

The authors declare the following competing interests: R.O., E.M. and C.K. are employees of Shimadzu Techno-research, Inc. S.-J.K. has personal financial interests as a share-holder in MedPacto Inc. All other authors have no relevant affiliations or financial involvement with any organization or entity with a financial interest in or financial conflict with the subject matter or materials discussed in the manuscript apart from those disclosed.
