## [Peer Review File · Nature Communications]

Reviewers' Comments:

Reviewer #1:

Remarks to the Author:

This is a review of lipidomics methods only

The description of methods is now far improved, but some issues remain to be sorted.

1. LysoPLs: the 20:4 and 22:6 (and maybe a few others?) aren't labelled on the chromatograms. If these are the large peaks that are unlabelled, can they label these.

A few of the peaks look to be below the LOD (which at 3 is right on the border, but ok). 16:1 is one of them, and the arrow seems to be pointing at baseline

For two others, there isn't a clear peak, it looks more like a smear, These are 20:2 and 20:3. Because they show them all in one graph they are very hard to see, and separating them would be better.

Often, with lysoPLs, there are two peaks, seen and in those cases, both should ideally be integrated to give a single value (see paper citation below), and this be stated. In fact, most lysoPL are detected as the 1-acyl-2-lyso forms, due to rapid spontaneous acyl migration (the second peak detected). Here it seems like only the first peak was integrated. They should consider this, and perhaps need to separate the chromatograms to show what was actually integrated, and also remove those that maybe below LOD

Pluckthun A and Dennis EA. Acyl and phosphoryl migration in lysophospholipids: importance in phospholipid synthesis and phospholipase specificity. *Biochemistry*. 1982;21:1743-50.

2. For oxylipins, instead of CAS numbers (or in addition?) MRM transitions used should be included. Also chromatograms don't include any time axis so as they are shown, they are not very clear. Also, there are many in the assay that appear to have not been detected. Can they remove those, only show ones that were detected, and include a time axis for each so retention time is given (or label them with the retention time if this is easier).

Reviewer #2:

Remarks to the Author:

In the manuscript entitled, "Gdpd3 is a lysophospholipase D enzyme that maintains chronic myelogenous leukaemia stem cells" by Naka et al. they propose that lysophospholipid metabolism is important for CML stem cells. They explore the importance for lipid metabolism in CML by depleting Gdpd3 using mouse models. Using RNA-seq analysis they examine Lgr4/Gpr48 as a downstream ligand of this pathway. The authors present compelling data on GDPD3 in CML and CML stem cells, but additional data would strengthen their claims as some of the mechanistic studies do not quite fit with the GDPD3 phenotype. Moreover, there are other concerns about the rationale for some of the experiments.

Main comments:

1. The authors first utilize a transgenic mouse model for CML but then switch to using the retroviral model. The rationale for this switch is really not clear as it would make sense to perform all the experiments using the transgenic model over the Rv model. It is understandable if the transgenic model is more challenging to work with but then this should be stated.

2. The authors state that there is no effect on normal stem cell function and that colony formation is normal. Can they provide this data on the normal mice such as WBC counts, multi-lineage colony

formation data (granulocyte, meg, erythroid mixed), replating info and enumeration of phenotypic HSC and HPCs? If they really want to claim CML-LIC specific phenotypes then, they should provide some transplant or serial transplant data.

3. The Gpdd3^{-/-} LT-CML cells displayed greater colony forming capacity than Gdpd3^{+/+} stem cells. What is their cell cycle status? Are they more proliferative and less quiescent and thus likely to exhaust?

4. In line with the increased colony forming capacity, the mice transplanted with Gdpd3^{-/-} retro CML LSK develop disease more rapidly than Gdpd3^{+/+}, can the authors show the % and the numbers of LT and ST in primary and secondary mice? Do Gdpd3 mice have already less LT-LSC in primary mice? Or they are just less functional?

5. The authors focus on lipid metabolism for this paper but a rationale for the genes chosen as promising candidates needs to be strengthened. Can the authors provide some data and rationale for choosing this pathway and also the downstream target mechanism.

6 The mechanism of the manuscript is not consistent with their hypothesis. If Lgr4 binds to the unknown downstream lipid products of GDPD3 than shouldn't there be a better correlation with the two phenotypes. Loss of Gdpd3 results in an increase in phenotypic CML-LSCs and a more rapid disease initially and then reduced leukemia activity in secondary transplants. If they want to prove that this is their mechanism an add back of Lgr4 rescuing the phenotype would be more appropriate. Additionally, if possible an add back of specific lipids would also strengthen their claims that these alternative lipid products could be more important. For example, prostaglandins have been implicated in controlling hematopoietic stem cell function in different contexts.

7. What is the phenotype of the Lgr4 depleted leukemic mice and what is the frequency of LT-LSCs?

8 Besides their mechanistic studies their human relevant data could also be strengthened. What is the level of Gdpd3 in CML compared with normal CD34 in healthy subjects? Can Gdpd3 expression be a predictive marker of TKI resistance? Does loss of Gdpd3 in human CML-BC cell lines have any effect on differentiation or engraftability?

Responses to Reviewers

Reviewer #1:

The description of methods is now far improved, but some issues remain to be sorted.

- 1-1) LysoPLs: the 20:4 and 22:6 (and maybe a few others?) aren't labelled on the chromatograms. If these are the large peaks that are unlabelled, can they label these. A few of the peaks look to be below the LOD (which at 3 is right on the border, but ok). 16:1 is one of them, and the arrow seems to be pointing at baseline. For two others, there isn't a clear peak, it looks more like a smear, These are 20:2 and 20:3. Because they show them all in one graph they are very hard to see, and separating them would be better.

Response 1-1)

We sincerely appreciate Reviewer #1's kind comment that our methods description has improved. We apologize that the chromatograms for some LysoPAs were difficult to distinguish in our original Supplementary Method 1. As suggested, we now separately indicate the chromatograms of each LysoPA in our revised Supplementary Method 1. On these chromatograms, we have labeled several LysoPA peaks (such as LPA20:4 and LPA22:6), which were not indicated in our original Supplementary Method 1. We have also included explanations of large peaks as "LPC" or "unidentified peak". We evaluated LOD using Analyst Ver. 1.7 (Sciex) software, and we hope that Reviewer #1 agrees to recognize some weak peaks (*i.e.*, LPA16:1, LPA20:2, and LPA20:3) as indicated in our revised Supplementary Method 1.

- 1-2) Often, with lysoPLs, there are two peaks, seen and in those cases, both should ideally be integrated to give a single value (see paper citation below), and this be stated. In fact, most lysoPL are detected as the 1-acyl-2-lyso forms, due to rapid spontaneous acyl migration (the second peak detected). Here it seems like only the

first peak was integrated. They should consider this, and perhaps need to separate the chromatograms to show what was actually integrated, and also remove those that maybe below LOD.

Pluckthun A and Dennis EA. Acyl and phosphoryl migration in lysophospholipids: importance in phospholipid synthesis and phospholipase specificity. *Biochemistry*. 1982;21:1743-50.

Response 1-2)

Following Reviewer #1's insightful suggestion, we did indeed detect some doublet peaks of LPA20:3, and LPA18:0 in the separated chromatograms now presented in revised Supplementary Method 1. We also detected multiple peaks of LPA18:1. We confirmed that we evaluated the "integrated area" of these LPAs in our original manuscript.

- 2-1) For oxylipins, instead of CAS numbers (or in addition?) MRM transitions used should be included.

Response 2-1)

We now indicate MRM transitions for all 88 lipid mediators detected in our experiments and also include CAS numbers in revised Supplementary Method 2. The MRM transitions are also indicated on the upper left of chromatograms of the lipid mediators in new Supplementary Method 3 (as described in Response 2-2 below).

Restrictions imposed by the independent company (Shimadzu inc. Kyoto, Japan) that we used to identify lipid mediators allowed us to present this information only for the lipid mediators actually detected in our experiments and the lipid mediators that were reported previously (Ref: 50) (total 151 MRM transitions). Thus, we could not obtain this information for mediators

that were not detected in our work. We hope that Reviewer #1 is satisfied with this explanation.

- 2-2) Also chromatograms don't include any time axis so as they are shown, they are not very clear. Also, there are many in the assay that appear to have not been detected. Can they remove those, only show ones that were detected, and include a time axis for each so retention time is given (or label them with the retention time if this is easier).

Response 2-2)

We would like to thank Reviewer #1 for his/her sensible suggestion that we indicate only chromatograms for lipid mediators that were detected. In new Supplementary Method 3, we present representative 88 chromatograms for lipid mediators detected and IS. On these chromatograms, we indicate Y-axis (absolute abundance), X-axis (time), MRM transition, and retention time. We are grateful to Reviewer #1 for helping us to greatly improve our description of our lipidomics methodology.

Reviewer #2:

In the manuscript entitled, “Gdpd3 is a lysophospholipase D enzyme that maintains chronic myelogenous leukaemia stem cells” by Naka et al. they propose that lysophospholipid metabolism is important for CML stem cells. They explore the importance for lipid metabolism in CML by depleting Gdpd3 using mouse models. Using RNA-seq analysis they examine Lgr4/Gpr48 as a downstream ligand of this pathway. The authors present compelling data on GDPD3 in CML and CML stem cells, but additional data would strengthen their claims as some of the mechanistic studies do not quite fit with the GDPD3 phenotype. Moreover, there are other concerns about the rationale for some of the experiments.

Main comments:

1. The authors first utilize a transgenic mouse model for CML but then switch to using the retroviral model. The rationale for this switch is really not clear as it would make sense to perform all the experiments using the transgenic model over the Rv model. It is understandable if the transgenic model is more challenging to work with but then this should be stated.

Response 1

First of all, we thank Reviewer #2 for his/her helpful suggestions to improve our manuscript. We apologize that our explanation for our use of two CML mouse models was not clear. We have expanded our description to explain the advantages of each CML model and why we chose them in our revised Results section on page 8, line 10.

As is now described in the revised Results, we believe that the tet-inducible CML model (tet-CML mice) is better-suited to evaluating the natural development of CML disease than is the retroviral CML model (retro-CML mice). In addition, it is easier more accurate to examine molecular mechanisms in the most primitive

LT-CML stem cells (such as applying fluorescent immunohistochemistry) using the tet-CML model. On the other hand, the retro-CML model is useful for examining the functionality (*i.e.*, CML disease-initiating capacity) of BCR-ABL1/EGFP-positive LSK cells; *i.e.*, isolate these cells by cell sorting and use them for serial bone marrow transplantation (BMT). However, it is difficult to detect the most primitive LT-CML stem cells in the retro-CML model.

In our revised manuscript, we now present the survival rates of *Gdpd3*^{+/+} tet-CML and *Gdpd3*^{-/-} tet-CML mice in revised Figure 1g. Interestingly, *Gdpd3*^{-/-} tet-CML mice developed CML disease more rapidly than *Gdpd3*^{+/+} tet-CML mice, *i.e.*, within 4 months of CML induction by Dox withdrawal. However, by 6 months post-Dox withdrawal, the disease-initiating capacity of *Gdpd3*^{-/-} tet-CML cells was greatly reduced compared to that of *Gdpd3*^{+/+} tet-CML cells. These results suggest that *Gdpd3*^{-/-} CML stem cells actively develop CML disease at first but then have a reduced ability to give rise to differentiated CML cells. Thus, to better examine the disease-initiating capacity of *Gdpd3*-deficient “CML stem/progenitor cells”, we employed serial transplantation of CML-LSK cells isolated from retro-CML mice (revised Figure 2). Throughout our revised manuscript, we have made sincere efforts to explain why one or the other mouse CML model was used, as recommended by Reviewer #2.

2. The authors state that there is no effect on normal stem cell function and that colony formation is normal. Can they provide this data on the normal mice such as WBC counts, multi-lineage colony formation data (granulocyte, meg, erythroid mixed), replating info and enumeration of phenotypic HSC and HPCs? If they really want to claim CML-LIC specific phenotypes then, they should provide some transplant or serial transplant data.

Response 2

We thank the reviewer for this comment and have added blood cell count data for normal *Gdpd3*^{+/+} and *Gdpd3*^{-/-} mice at 10 and 39 weeks of age (revised Supplementary Table 1). Although the red blood cell (RBC) count and hematocrit (HCT) values were slightly higher in normal *Gdpd3*^{-/-} mice than in normal *Gdpd3*^{+/+} mice, no difference was observed in white blood cell (WBC) count. We have also added data on multi-lineage colony formation (CFU-GM, BFU-E, and CFU-Mix) to revised Supplementary Figure 4. These results all indicate that hematopoiesis, in general, not affected by *Gdpd3* deficiency.

We also followed Reviewer #2's insightful suggestion and have performed serial transplantation experiments using normal hematopoietic stem/progenitor cells (LSK cells). Bone marrow reconstitution capacity was slightly higher for *Gdpd3*^{-/-} LSK cells than for *Gdpd3*^{+/+} LSK cells during a first-round of bone marrow transplantation (BMT), as indicated in revised Supplementary Figure 5a. Similarly, *Gdpd3*^{-/-} LSK cells maintained their BM reconstitution capacity after a second-round of BMT, whereas the capacity of *Gdpd3*^{+/+} LSK cells was decreased (revised Supplementary Figure 5b). Thus, *Gdpd3*-deficient normal hematopoietic stem/progenitor cells (LSK cells) maintain their self-renewal capacity at least through two rounds of serial BMT *in vivo*. Because the CML disease-initiating capacity of *Gdpd3*-deficient CML stem/progenitor cells (LSK cells) was attenuated during a second-BMT (original Figure 1g, revised Figure 2b), our results indicate that *Gdpd3* is required specifically to maintain the self-renewal capacity of CML stem cells rather than that of normal hematopoietic stem cells (HSCs) *in vivo*. We describe these new data in the revised Results section on page 10, line 5.

Our findings suggest that, despite their expression of the *BCR-ABL1* oncogene, CML stem cells may require *Gdpd3*-mediated lysophospholipid metabolism to maintain their stemness *in vivo*, and that this support occurs in an oncogene-independent manner. We now explain our understanding of *Gdpd3*'s biological significance specific to CML stem cells in our revised Discussion from page 23, line 18 to page 24, line 10.

3. The *Gpd3*^{-/-} LT-CML cells displayed greater colony forming capacity than *Gpd3*^{+/+} stem cells. What is their cell cycle status? Are they more proliferative and less quiescent and thus likely to exhaust?

Response 3

We thank the reviewer for this valuable question. We examined the cell cycle distribution of *Gdpd3*^{+/+} and *Gdpd3*^{-/-} CML-LSK cells in our retro-CML mice using BrdU incorporation assays *in vivo*. The frequency of BrdU⁺ cells was dramatically elevated among splenic *Gdpd3*^{-/-} CML-LSK cells compared to splenic *Gdpd3*^{+/+} CML-LSK cells (new Figure 2g, h). Consistent with this finding, the frequency of BrdU⁻ G₀/G₁ cells was reduced among *Gdpd3*^{-/-} CML-LSK cells compared to *Gdpd3*^{+/+} CML-LSK cells. The increased frequency of splenic BrdU⁺ proliferative cells explains the elevations in absolute number of *Gdpd3*^{-/-} CML-LSK cells compared with *Gdpd3*^{+/+} CML-LSK cells (as also described in Response 4 below). The frequency of BrdU⁺ cells also appeared to be higher among BM *Gdpd3*^{-/-} CML-LSK cells compared to *Gdpd3*^{+/+} CML-LSK cells (new Supplementary Figure 7). These results demonstrate that *Gdpd3* deficiency activates the growth of CML stem/progenitor cells, breaking the quiescence that CML stem cells require to sustain their “stemness” *in vivo*. Thus, our findings support our hypothesis that *Gdpd3*-mediated lysophospholipid metabolism plays an essential role in oncogene-independent maintenance of CML stem cells. We describe these new data in the revised Results on page 12, line 10.

4. In line with the increased colony forming capacity, the mice transplanted with *Gdpd3*^{-/-} retro CML LSK develop disease more rapidly than *Gdpd3*^{+/+}, can the authors show the % and the numbers of LT and ST in primary and secondary mice? Do *Gdpd3* mice have already less LT-LSC in primary mice? Or they are just less functional?

Response 4

We agree with Reviewer #2 that we should clarify whether *Gdpd3* deficiency decreases the number or functionality of CML stem cells *in vivo*. To address this issue, we determined the absolute number and frequency of BCR-ABL1/EGFP positive CML LSK cells after first- and second-rounds of serial BMT. As explained in Response 1 above, it is technically very difficult to evaluate “LT-CML stem cells” in the retro-CML model.

In BM, no significant differences were observed in either the frequency or absolute number of BCR-ABL1/EGFP positive CML-LSK cells after the first-BMT (new Figure 2c; new Supplementary Figure 6a). In spleen, consistent with their activated cell division (Response 3), the absolute number of *Gdpd3*^{-/-} CML-LSK cells was dramatically increased after the first-BMT, although no difference in frequency was detected (new Figure 2d; new Supplementary Figure 6b). This elevated absolute number of *Gdpd3*^{-/-} CML-LSK cells in spleen was due to a higher number of total splenocytes in *Gdpd3*^{-/-} retro-CML mice compared to *Gdpd3*^{+/+} retro-CML mice (data not shown). However, to our surprise, the frequency and absolute number of *Gdpd3*^{-/-} CML-LSK cells were significantly decreased after a second-round of BMT both in BM and spleen of recipients (new Figure 2e, f; new Supplementary Figure 6c, d).

As indicated in original Figure 1f and revised Figure 2a, recipients bearing *Gdpd3*^{-/-} CML-LSK cells developed CML disease more rapidly than recipients bearing *Gdpd3*^{+/+} CML-LSK cells after the first-round of BMT. However, after the second-round, CML disease developed more slowly in recipients bearing *Gdpd3*^{-/-} CML-LSK cells than in those bearing *Gdpd3*^{+/+} CML-LSK cells (original Figure 1g; revised Figure 2b). These results clearly indicate that it is the exhaustion of CML-LSK cells, rather than loss of function, that decreases the development of CML disease after the second-BMT. We have described these new data in the revised Results from page 11, line 9 to page 12, line 10.

- 5-1. The authors focus on lipid metabolism for this paper but a rationale for the genes chosen as promising candidates needs to be strengthened.

Response 5-1

We apologize for our inadequate description of our rationale for choosing to study the *Gdpd3* gene and lysophospholipid metabolism, and agree with Reviewer #2 that a better explanation is warranted.

In 2010, we reported that the forkhead O transcription factor FoxO3a plays an essential role in the maintenance of CML stem cells (Naka *et al.*, Nature 2010). Although TGF- β signaling is known to regulate Foxo3a's function, it was not understood how CML stem cells suppress the PI3K/AKT pathway, which is regulated by lipid metabolism. Moreover, this suppression appears to occur in an oncogene-independent manner. In 2015, we addressed this fundamental question by using RNA sequencing to compare the mRNA expression patterns of LT-CML stem cells and normal HSCs (GSE 70031, Naka *et al.*, Nature Communications 2015, DOI: 10.1038/ncomms9039). Among the genes that differed between these two cell populations and was involved in lipid metabolism was *Gdpd3*.

In the present study, we described that *Gdpd3* mRNA was highly expressed in LT-CML stem cells compared to normal LT-HSCs by RNA sequencing (GSE 70031) (new Supplementary Figure 1). We next examined *Gdpd3* mRNA levels by qRT-PCR and confirmed that they were higher in LT-CML stem cells than in normal LT-HSCs (revised Figure 1b). Notably, *Gdpd3* mRNA was increased in CML LT stem cells and CML CD48⁺ LSK cells, but not in CML MPP (multipotent progenitors) or CML LK cells. Transduction of two different siRNAs targeting mouse *Gdpd3* mRNA suppressed the colony-forming capacity of CML-LSK cells (new Figure 1c). These results reinforced our determination to find out if *Gdpd3* and lysophospholipid metabolism play important roles in maintaining primitive CML stem cells. Thus, we established *Gdpd3*-deficient mice by genome-editing to examine the role of *Gdpd3* in CML stem cells *in vivo*. Indeed,

our data suggested that CML BM cells and CML-LSK cells showed reduced lysophospholipid metabolism (original Figure 3a; revised Figure 3a; new Figure 3b). We include these new data and our expanded rationale in the revised Results from page 8, line 3 to page 9, line 16.

- 5-2. Can the authors provide some data and rationale for choosing this pathway and also the downstream target mechanism.

Response 5-2

As described in Response 5-1 above, we previously investigated lipid metabolism upstream of Foxo3a in mouse LT-CML cells and identified *Gdpd3* as a gene of interest by RNA sequencing.

To explore the downstream target mechanism, we employed three approaches:

1) We used lipidomics to determine LPA levels in CML-LSK cells, since LPAs are a direct target of Gdpd3 lysophospholipase D activity; 2) We measured levels of phosphorylated AKT and S6 ribosomal protein, which reflect activation of the AKT/mTORC1 pathway; and 3) We monitored the subcellular localization of Foxo3a by fluorescence immunohistochemistry (FIHC).

Because Gdpd3 is a lysophospholipase D enzyme, we first examined LPA levels in BM mononuclear cells (original Figure 3a; revised Figure 3a), followed by CML-LSK cells (new Figure 3b). Importantly, *Gdpd3* deficiency appeared to decrease the levels of certain LPAs, such as LPA18:1 and LPA 18:0, both in CML BM cells and CML-LSK cells. These results indicated that Gdpd3's lysophospholipase D activity is crucial for producing and/or recycling LPAs. We describe these data in revised Results on page 13, line 11.

We next examined levels of phosphorylated AKT and S6 ribosomal protein, as well as the subcellular localization of Foxo3a. Phospho-AKT and phospho-S6 were both increased in *Gdpd3*^{-/-} LT-CML stem cells compared to *Gdpd3*^{+/+} LT-CML

stem cells, indicating that Gdpd3 suppresses the AKT/mTORC1 pathway in LT-CML stem cells (new Figure 5a,b; new Supplementary Figure 8a,b). More importantly, whereas Foxo3a was located within the nuclei of *Gdpd3*^{+/+} LT-CML stem cells, it was predominantly detected in the cytoplasm of *Gdpd3*^{-/-} LT-CML stem cells (new Figure 5c; new Supplementary Figure 8c). These results suggested that *Gdpd3* deficiency activates the AKT/mTORC1 pathway and thereby inhibits the nuclear localization of Foxo3a, resulting in decreased maintenance of CML stem cells *in vivo*.

Collectively, our new data indicate that Gdpd3 acts as a key regulator of lysophospholipid metabolism in CML-LSK cells. Despite their expression of the *BCR-ABL1* oncogene, Gdpd3-expressing CML stem cells can suppress the AKT/mTORC1 pathway and thus their proliferation (Response 3). Due to this repression of AKT activity, Foxo3a translocates into the nucleus and contributes to the maintenance of CML stemness in an oncogene-independent manner. We present this hypothesis in the revised Abstract, in the revised Discussion from page 23, line 18 to page 24, line 10 (Response 2), and in new Figure 8c. We sincerely appreciate Reviewer #2's helpful comments, which have driven us to better understand CML stem cell biology and greatly improve our paper.

- 6-1. The mechanism of the manuscript is not consistent with their hypothesis. If Lgr4 binds to the unknown downstream lipid products of GDPD3 than shouldn't there be a better correlation with the two phenotypes. Loss of Gdpd3 results in an increase in phenotypic CML-LSCs and a more rapid disease initially and then reduced leukemia activity in secondary transplants. If they want to prove that this is their mechanism an add back of Lgr4 rescuing the phenotype would be more appropriate.

Response 6-1

We understand the reviewer's concern. We think it is possible that, although Lgr4/Gpr48 is definitely important for CML stem cell maintenance, it is not the

only critical factor, and that changes to multiple pathways may contribute to the defective self-renewal capacity of *Gdpd3*^{-/-} CML stem cells. We cite two lines of evidence to support this position. Firstly, when we carried out gene expression profiling by RNA-sequencing, we found that several GPCR (G-protein coupled receptor) genes (not only *Lgr4*/*Gpr48*) were decreased in *Gdpd3*^{-/-} LT-CML stem cells compared to *Gdpd3*^{+/+} CML stem cells (original Figure 4a; original Supplementary Figure 3; revised Figure 7a; revised Supplementary Figure 11). Secondly, *Gdpd3* deficiency resulted in an increase in AKT/mTORC1 signaling (Response 5-2) as well as loss of interaction between Foxo3a and active β -catenin (see Response 6-2 below). Moreover, enforced treatment with PGE₂ could not restore the binding between Foxo3a and active β -catenin in *Gdpd3*^{-/-} CML stem cells. Thus, we concluded that *Gdpd3*-mediated lysophospholipid metabolism might affect multiple pathways contributing to the maintenance of CML stemness. Because it became apparent that more than one factor was likely involved, we concluded it was logical to abandon our experiment intended to rescue the *Lgr4* gene. We describe this train of thought in our revised Discussion from page 25, line 18 to page 26, line 11 and in new Figure 8c. We hope that Reviewer #2 agrees with us that such a rescue experiment would not be particularly helpful under the current circumstances.

Nevertheless, because we could not determine whether transduction of *Lgr4* cDNA would truly rescue *Gdpd3* deficiency, we have removed all sentences in our revised paper referring to “an axis between *Gdpd3* and *Lgr4*”.

- 6-2. Additionally, if possible an add back of specific lipids would also strengthen their claims that these alternative lipid products could be more important. For example, prostaglandins have been implicated in controlling hematopoietic stem cell function in different contexts.

Response 6-2

We thank Reviewer #2 for this insightful suggestion. It was previously reported that PGE₂ activates β -catenin via EP1 in CML stem cells (Ref. 24, Li *et al.*, Cell Stem Cell 2017), and we found that PGE₂ was decreased in *Gdpd3*^{-/-} CML BM cells compared to *Gdpd3*^{+/+} CML BM cells (original Figure 3b; revised Figure 4a). It was also known that Foxo3a interacts with β -catenin in metastatic colon cancer cells (new Ref. 41, Tenbaum *et al.*, Nat Med 2012). In our study, we used highly sensitive Duolink[®] *in situ* PLA technology to examine the effect of PGE₂ treatment on the interaction between Foxo3a and β -catenin in LT-CML stem cells.

We confirmed the interaction of Foxo3a with active β -catenin within the nuclei of WT LT-CML stem cells (new Figure 8a; new Supplementary Figure 14). Interestingly, however, we could not detect Foxo3a/ β -catenin binding within the nuclei of LT-CML stem cells isolated from either *Gdpd3*^{-/-} tet-CML mice or *Lgr4*^{Gt/Gt} tet-CML mice (new Figure 8a; new Supplementary Figure 14). When we treated WT LT-CML stem cells with PGE₂, we saw no obvious effects on Foxo3a/ β -catenin binding (new Figure 8b; new Supplementary Figure 15). Treatment of *Gdpd3*^{-/-} LT-CML stem cells with PGE₂ *in vitro* only slightly increased this interaction in the nucleus. In contrast, treatment of *Lgr4*^{Gt/Gt} tet-CML stem cells *in vitro* with PGE₂ dramatically increased the interaction of Foxo3a and active β -catenin in the nucleus. These results suggested that PGE₂ treatment triggered distinct effects in *Gdpd3*^{-/-} LT-CML stem cells and *Lgr4*^{Gt/Gt} LT-CML stem cells such that PGE₂ was able to rescue Foxo3a/ β -catenin binding in *Lgr4*^{Gt/Gt} LT-CML stem cells but not in *Gdpd3*^{-/-} LT-CML stem cells. This difference in response to PGE₂ is probably due to differences in types or levels of intracellular lipid components. *Gdpd3*^{-/-} CML stem cells show decreased production of several lysophospholipids and lipid mediators, as well as altered downstream AKT signaling that impairs Foxo3a/ β -catenin binding. By contrast, because the only defect in *Lgr4*^{Gt/Gt} CML stem cells is decreased expression of *Lgr4* mRNA, PGE₂ treatment was able to restore Foxo3a/ β -catenin binding (see Figure 8c).

Collectively, these new data inspired by the comments of Reviewer #2 indicate that PGE₂ production plays a vital role in activating β -catenin to regulate its binding to Foxo3a in LT-CML stem cells. These data are described in the revised Results from page 21, line 1 to page 22, line 12, the revised Discussion on page 25, line 18, and in new Figure 8c.

7. What is the phenotype of the *Lgr4* depleted leukemic mice and what is the frequency of LT-LSCs?

Response 7

To address this question, we examined the frequency and absolute number of LT-CML stem cells in our *Lgr4*^{Gt/Gt} tet-CML mouse model, and the frequency and absolute number of BCR-ABL1/EGFP⁺ LSK cells in our *Lgr4*^{Gt/Gt} retro-CML mouse model. Absolute numbers of LT-CML stem cells were comparable between *Lgr4*^{+/+} tet-CML and *Lgr4*^{Gt/Gt} tet-CML mice, whereas the frequency of these cells appeared to be increased in *Lgr4*^{Gt/Gt} tet-CML mice (new Figure 7c; new Supplementary Figure 13a). Consistent with this finding, we saw similar frequencies and absolute numbers of BCR-ABL1/EGFP⁺ LSK cells in *Lgr4*^{+/+} retro-CML and *Lgr4*^{Gt/Gt} retro-CML mice at 20 days post-transplantation, but a significant decrease in BCR-ABL1/EGFP⁺ LSK cells in *Lgr4*^{Gt/Gt} retro-CML mice by 90 days post-transplantation (new Figure 7f; new Supplementary Figure 13b). In the same vein, the frequency of BCR-ABL1/EGFP⁺ mature CML cells was significantly decreased in *Lgr4*^{Gt/Gt} retro-CML mice compared with *Lgr4*^{+/+} retro-CML mice (new Figure 7g). Thus, whereas *Lgr4* is dispensable for the development of CML stem/progenitor cells, it plays an important role in the long-term maintenance of CML-initiating stem/progenitor cells. We describe these data in the revised Results from page 19, line 15 to page 20, line 17.

8-1. Besides their mechanistic studies their human relevant data could also be strengthened. What is the level of Gdpd3 in CML compared with normal CD34 in healthy subjects?

Response 8-1

To address this question, we investigated expression levels of GDPD3 mRNA in hematopoietic cells of healthy individuals and CML patients as recorded in a public database (please see Figure for Reviewer #2 below). Unexpectedly, we could not detect a difference in GDPD3 mRNA expression in $\text{Lin}^- \text{CD34}^+ \text{CD38}^- \text{CD90}^+$ hematopoietic stem cells of these individuals.

It is possible that, unlike mice, the expression of the *GDPD3* gene is not increased in human CML cases. Another possibility is that the CML stem cell populations assessed in this database include both Philadelphia chromosome⁺ (Ph⁺) cells and Ph⁻ cells. In an attempt to resolve this issue, we transduced siRNAs targeting human *GDPD3* mRNA into the K562 human CML cell line, as well as into primary human CD34⁺ CML cells isolated from a chronic phase CML patient, and examined colony-forming capacity *in vitro*. Whereas the transduction of *GDPD3* siRNA only weakly suppressed the colony-forming capacity of K562 cells, it robustly reduced the colony-forming capacity of human CD34⁺ CML cells (revised Fig.6e,f). These results suggest that, in the human context, *GDPD3* may make a greater contribution to the maintenance of primitive CML cells rather than to the support of mature differentiated CML cells. We describe these data in the revised Results from page 17, line 14 to page 18, line 4. We continue our investigations of this issue and intend to report our findings in a future publication.

8-2. Can *Gdpd3* expression be a predictive marker of TKI resistance?

Response 8-2

We thank the reviewer for this intriguing question. To investigate the role of *GDPD3* in TKI resistance in human CML stem cells, we transduced human CD34⁺ CML cells with *GDPD3* siRNA in the presence/absence of the TKI imatinib. We found that repression of *GDPD3* mRNA effectively suppressed colony formation by imatinib-resistant CD34⁺ CML cells (revised Fig.6g). These results suggest that *GDPD3* may be involved in promoting the survival of primitive CML stem cells even in the presence of TKI, a hypothesis we describe in our revised Results on page 18, line 4-10. We are currently interested whether determination of levels of *GDPD3*

mRNA in bone marrow CD34⁺ CML cells combined with measurement of lipid metabolites such as LPAs or other lipid mediators in peripheral blood can facilitate early diagnosis of TKI resistance in CML patients. Our future research plan is focused on assessing the predictive value of these potential markers emerging from our work.

8-3. Does loss of Gdpd3 in human CML-BC cell lines have any effect on differentiation or engraftability?

Response 8-3

To address Reviewer #2's concern, we conducted a preliminary experiment in which murine CML-LSK cells were transduced with shRNA via EGFP-labeled lentiviral vector and transplanted into the BM of syngeneic recipients. Unexpectedly, however, the *in vivo* expression of shRNA as monitored by EGFP seemed to disappear due to an unknown gene silencing mechanism. Thus, we regret to report that we could not examine the effect of loss of GDPD3 on the differentiation or engraftability of human CML cells *in vivo*. Instead, as detailed in Responses 8-1 and 8-2 above, we transduced siRNA targeting GDPD3 mRNA into primary human CD34⁺ CML cells *in vitro* and observed suppression of colony formation. Once again, we truly appreciate Reviewer #2's logical and sincere advice, which has helped us to greatly improve our manuscript.

Corrections initiated by the authors:

We have made the following additional changes to our revised manuscript.

To better reflect our newly obtained results, we would like to slightly change the title of our manuscript from the original title “Gdpd3 is a lysophospholipase D enzyme that maintains chronic myelogenous leukaemia stem cells” to “The lysophospholipase D enzyme Gdpd3 is required to maintain chronic myelogenous leukaemia stem cells”.

The word count for our revised Abstract has been reduced to the required 150 words.

In our revised Introduction on page 5, line 12, we have added the sentence “Despite their expression of the *BCR-ABL1* oncogene, CML stem cells have been reported to maintain their “stemness” in an oncogene-independent manner, the mechanism of this maintenance is unknown” and cited a reference 18 (Corbin *et al.*, J Clin. Invest. 2011). This addition provides a rationale for the new results presented in Figure 2g,h and Figure 5a-c that were obtained during the revision period as we acted on the reviewers’ valuable comments.

We indicated LPAs as “LPA C18:1” in original manuscript. However, it is not clear that “LPA C18:1” represents unsaturated alkoxy group or unsaturated carboxyl group. To be clear, we now indicate such LPAs using an “LPA18:1” format throughout our revised manuscript.

During the revision period, we examined LPA levels in *Gdpd3*^{+/+} CML stem/progenitor (LSK) cells and in *Gdpd3*^{-/-} LT-CML cells. We obtained results that paralleled our original data obtained from total bone marrow cells (original Figure 3a). We have added the results of our new LPA determinations as new Figure 3b. Based on these new data, we have corrected two sentences as follows:

Original: “However, unexpectedly, there were no obvious differences in LPA levels between *Gdpd3*^{+/+} tet-CML cells and *Gdpd3*^{-/-} tet-CML cells. Thus, the exact nature of the products of Gdpd3 enzymatic activity in the CML context is unclear.” in the original Results on page 13, line 4.

Corrected: “In contrast, most of these same LPAs tended to be decreased in both BMMNCs and LSK cells isolated from *Gdpd3*^{-/-} tet-CML mice compared to those from *Gdpd3*^{+/+} tet-CML mice (Fig.3a,b; Source Data Fig.3). Thus, in general, loss of Gdpd3 enzymatic activity decreases LPA levels in the CML context.” in the revised Results on page 13, line 11.

To reflect these newly obtained results, we have largely re-written our Discussion in our revised manuscript. We have also included additional methods (competitive repopulation assay, RNA interference in human CML CD34⁺ cells, cell cycle analysis, fluorescence immunostaining, and Duolink[®] *in situ* PLA) in our revised Methods section. We have removed 2 references (Ref. 41 and 42 in original References) but have added 6 new and crucial references (Ref 18, Corbin *et al.*, Ref. 37, Carmon *et al.*, Ref. 38, de Lau *et al.*, Ref. 39, Glinka *et al.*, Ref., 40 Heidel *et al.*, and Ref. 41 Tenbaum *et al.* to our revised References section).

For our mouse survival data, we have prolonged the observation period of WT retro-CML mice and *Lgr4*^{Gt/Gt}-retro-CML mice from 60 days (original Figure 4d) to 90 days (revised Figure 7e).

We have corrected some inaccurate CAS numbers of enantiomers of lipid compounds in revised Supplementary Method 2.

We have added three authors to our revised manuscript: Dr. Y. Sotomaru, who supported animal experiments during the revision period; and Drs. K. Mitani and K. Sasaki, who provided a human CML sample essential for addressing the reviewer’s

queries.

In the revised Acknowledgements, we have added information on a grant awarded to Dr. K. Naka.

We have deposited RNA-sequencing data of *Gdpd3*^{+/+} and *Gdpd3*^{-/-}LT-CML stem cells into a public database gene expression omnibus (GEO, ID: GSE149442) in NCBI, NIH, USA (<https://www.ncbi.nlm.nih.gov/gds/>).

Reviewers' Comments:

Reviewer #1:

Remarks to the Author:

Thanks to the authors for addressing my comments and the MS data for the lysoPAs is much improved. I would say however that the data for 24:1 and 24:0 is not high quality and maybe better removed from the dataset. Specifically, given the peak identified is co-eluting with another lipid, it would be extremely difficult to integrate this properly.

In relation to oxylipin data, a large panel of chromatograms are now presented, and I am grateful for this level of detail as it is very helpful. This sort of raw data is very good to show like this in supplementary. Out of the large number shown (88) most are fine however 11 could be considered somewhat problematic and either more evidence for structure is needed or they should be removed from the dataset. The specific problems with these are either (i) signal:noise is low, meaning they are not at least 5 times the height of the surrounding baseline noise, and so they are below the general LOQ used by researchers in the field, or (ii) they show split peaks or strong co-elution with nearby peaks. In some cases, two peaks are integrated so it is not clear which is the one of interest or why two peaks were integrated. Here are the lipids and it is important all of these are carefully checked and either verified using MS/MS or other methods, or removed. TXB1, 5S,14R-LXB4, LTC4, LTD4, RvD5, 9,10-diHOME, 12-keto-LTB4, 9-HPODE, Az-PAF, 15-KEDE, EPA

Reviewer #2:

Remarks to the Author:

The authors have responded to all the critiques.

Responses to Reviewer

Reviewer #1 (Remarks to the Author):

- 1-1) Thanks to the authors for addressing my comments and the MS data for the lysoPAs is much improved. I would say however that the data for 24:1 and 24:0 is not high quality and maybe better removed from the dataset. Specifically, given the peak identified is co-eluting with another lipid, it would be extremely difficult to integrate this properly.

Response 1-1)

We greatly appreciate Reviewer #1's insightful comments on our first revised manuscript and his/her help in significantly improving our paper. As requested, we have now deleted the chromatograms of the LysoPAs 24:1 and 24:0 from the second revised Supplementary Method3. We have also deleted the data of the LysoPAs 24:1 and 24:0 from the second revised Source Data Figure 3.xlsx.

- 1-2) In relation to oxylipin data, a large panel of chromatograms are now presented, and I am grateful for this level of detail as it is very helpful. This sort of raw data is very good to show like this in supplementary. Out of the large number shown (88) most are fine however 11 could be considered somewhat problematic and either more evidence for structure is needed or they should be removed from the dataset. The specific problems with these are either (i) signal:noise is low, meaning they are not at least 5 times the height of the surrounding baseline noise, and so they are below the general LOQ used by researchers in the field, or (ii) they show split peaks or strong co-elution with nearby peaks. In some cases, two peaks are integrated so it is not clear which is the one of interest or why two peaks were integrated. Here are the lipids and it is important all of these are carefully checked and either verified using MS/MS or other methods, or removed.

TXB1, 5S,14R-LXB4, LTC4, LTD4, RvD5, 9,10-diHOME, 12-keto-LTB4, 9-HPODE, Az-PAF, 15-KEDE, EPA.

Response 1-2)

We have understood and agreed with Reviewer #1's kind advice. As requested, we have deleted the 11 oxylipins, TXB1 (37), 5S,14R-LXB4 (61), LTC4 (84), LTD4 (87), RvD5 (113), 9,10-diHOME (117), 12-keto-LTB4 (118), 9-HPODE (161), Az-PAF (198), 15-KEDE (208), and EPA (211) from the second revised Supplementary Method3. In the same vein, we would like to delete 7 oxylipins, TXB3 (26), 8,15-DiHETE (103), 10,17-DiHDHA (111), 12,13-DiHOME (116), 13-HDHA (168), 11-HDHA (177), and 9-HETE (179), which are also difficult to distinguish, from the second revised Supplementary Method3. We have deleted the bar graph of Az-PAF on the second revised Figure 4c. We have also omitted the values of the 11+7 oxylipins from the second revised Source Data Figure 4.xlsx. Once again, we sincerely appreciate that Reviewer #1 dedicatedly provided us valuable advice.

Reviewer #2 (Remarks to the Author):

The authors have responded to all the critiques.

Response to Reviewer #2:

We thank Reviewer #2 for his/her very helpful advice during our first round of revisions of our manuscript. We appreciate that Reviewer #2 agrees that we have fully addressed his/her valuable critiques.

Corrections initiated by the authours:

We have made the following changes to our second revised manuscript.

Original: “lysophospholipids (LPAs)” in the first revised Abstract.

Corrected: “lysophosphatidic acids” in the second revised Abstract in page 3.

We have made the following additional description and citation to our second revised manuscript in page 44, line 5.

“Note added in proof: During the reviewing process, a paper demonstrating that Lgr4 is essential for self-renewal in acute myelogenous leukaemia (AML) stem cells was published⁵³. The study supports our conclusion about the role of Lgr4 in maintenance of CML stem cells.”

- 53. Salik, B., *et al.* Targeting RSPO3-LGR4 signaling for leukemia stem cell eradication in acute myeloid leukemia. *Cancer Cell* 30, <https://doi.org/10.1016/j.ccell.2020.05.014> (2020).**